

# Archosauriform footprints in the Lower Triassic of Western Alps and their role in understanding the effects of the Permian-Triassic hyperthermal

Fabio Massimo Petti[1], Heinz Furrer[2], Enrico Collo[3], Edoardo Martinetto[4], Massimo Bernardi[1], Massimo Delfino[4,5], Marco Romano[6] and Michele Piazza[7]

[1] MUSE—Museo delle Scienze di Trento, Trento, Italy
[2] Paläontologisches Institut und Museum, Universität Zürich, Zürich, Switzerland
[3] Natura Occitana, Dronero (CN), Italy
[4] Dipartimento di Scienze della Terra, Università degli Studi di Torino, Turin, Italy
[5] Institut Català de Paleontologia Miquel Crusafont, Universitat Autónoma de Barcelona. Edifici ICTA-ICP, Barcelona, Spain
[6] Dipartimento di Scienze della Terra, Sapienza, University of Rome, Rome, Italy
[7] Dipartimento di Scienze della Terra, dell'Ambiente e della Vita, Università di Genova, Genoa, Italy

Corresponding author
Marco Romano,
marco.romano@uniroma1.it

## ABSTRACT

The most accepted killing model for the Permian-Triassic mass extinction (PTME) postulates that massive volcanic eruption (i.e., the Siberian Traps Large Igneous Province) led to geologically rapid global warming, acid rain and ocean anoxia. On land, habitable zones were drastically reduced, due to the combined effects of heating, drought and acid rains. This hyperthermal had severe effects also on the paleobiogeography of several groups of organisms. Among those, the tetrapods, whose geographical distribution across the end-Permian mass extinction (EPME) was the subject of controversy in a number of recent papers. We here describe and interpret a new Early Triassic (?Olenekian) archosauriform track assemblage from the Gardetta Plateau (Briançonnais, Western Alps, Italy) which, at the Permian-Triassic boundary, was placed at about 11° North. The tracks, both arranged in trackways and documented by single, well-preserved imprints, are assigned to *Isochirotherium gardettensis* ichnosp. nov., and are here interpreted as produced by a non-archosaurian archosauriform (erytrosuchid?) trackmaker. This new discovery provides further evidence for the presence of archosauriformes at low latitudes during the Early Triassic epoch, supporting a model in which the PTME did not completely vacate low-latitude lands from tetrapods that therefore would have been able to cope with the extreme hot temperatures of Pangaea mainland.

## INTRODUCTION

The Permian-Triassic mass extinction (PTME) was the most severe biotic crisis of all times (*Erwin, 1993*), eliminating >90% of marine and terrestrial species (*Erwin, 1993*; *Song et al.,*

*2013*; *Song et al., 2015*; *Romano et al., 2020*). After the mass extinction, totally new clades emerged, which include decapods and marine reptiles in the oceans and new tetrapods on land (*Chen & Benton, 2012*). In the last decade different physical environmental shocks have been identified as possible triggers for the huge crisis, which include increased atmospheric $CO_2$ concentrations, global warming, acid rain, ocean anoxia, ocean acidification and hypercapnia (*Chen & Benton, 2012*; *Benton, 2018*). The most accepted killing model (e.g., *Benton & Twitchett, 2003*; *Chen & Benton, 2012*; *Benton & Newell, 2014*; *Shen et al., 2019*) postulates an initial megascale eruption (more than 1,000 Gigatonnes of erupted lava, see *Grasby, Sanei & Beauchamp, 2011*), that released sulphate aerosols and methane from clathrate reservoirs (see *Berner, 2002*), which led to global warming and acid rain, causing a generalized plant die-offs and thus intensive erosion of the soil (*Wignall, 2001*; *Benton, 2003*; *Benton, 2018*; *Benton & Twitchett, 2003*; *Sephton et al., 2005*; *Knoll et al., 2007*). On land, habitable zones were drastically reduced, due to the combination of extreme heat, drought and acid rains, which caused progressive loss of soil and forests and had direct impact on lacustrine organisms and any land-dwelling animal (*Benton & Newell, 2014*).

According to several authors (*Joachimski et al., 2012*; *Sun et al., 2012*; *Schobben et al., 2014*; *Song et al., 2015*) the intense global warming started at the extinction horizon as testified in the Meishan section (South China). The process continued in the Early Triassic, very likely with the release of methane from deep ocean sediments and coals that triggered the process, and the release of additional greenhouse gases by interactions of the Siberian traps with local permafrost soils, limestones, and other deposits rich in organic matter (e.g., *Racki, 2003*; *Racki & Wignall, 2005*; *Retallack & Jahren, 2008*; *Grasby, Sanei & Beauchamp, 2011*).

The hyperthermal had severe effects also on the paleobiogeographic patterns. In recent years the distribution of land tetrapods across the PTME was discussed in a number of studies which however suggested different scenarios. By compiling literature evidence on the main skeletal findings, *Sun et al. (2012)* suggested that, in the Early Triassic, terrestrial vertebrates totally vacated the equatorial belt, the so-called 'vertebrate equatorial-gap', as a consequence of the extreme hot temperatures. More recently, *Bernardi et al. (2015)*, *Bernardi, Petti & Benton (2018)* and *Romano et al. (2020)* reviewed the late Permian-Early Triassic terrestrial tetrapod record integrating skeletal and track data and concluded that tetrapod geographic distribution was much wider than previously suggested. In the Early Triassic it included also the low latitudes, though polarward dispersals were detected in the Early Triassic and possibly linked to the development of super-hot temperatures in the equatorial belt (*Bernardi, Petti & Benton, 2018*). Fossil track evidence, in particular, was key in denying the existence of an 'equatorial gap' (*Bernardi, Petti & Benton, 2018*; *Romano et al., 2020*).

Archosaur tracks and trackways are in fact well-known from Lower to Middle Triassic siliciclastic and carbonate sediments of the Upper Buntsandstein and Lower Muschelkalk (late Olenekian-early Anisian) of Germany (*Haubold, 1971a*; *Haubold, 1971b*; *Klein & Haubold, 2007*), the Lower Triassic of the Holy Cross Mountains in Poland (*Klein & Niedźwiedzki, 2012*), the Middle Triassic of Bourgogne (*Gand, 1979*), Massif Central (*Demathieu, 1970*) and Ardèche in France (*Courel & Demathieu, 1976*), the Iberian Range

in Spain (*Fortuny et al., 2011*; *Dìaz-Martinez et al., 2015*) and Sardinia in Italy (*Citton et al., 2020*). Further sites, often with identical ichnotaxa and ichnoassemblages, are known from the Lower to Middle Triassic of Great Britain (*King et al., 2005*), North American southwest (*Klein & Lucas, 2010a*; *Klein & Lucas, 2010b*), Argentina (*Melchor & De Valais, 2006*), Africa (*Klein et al., 2011*) and southern China (*Xing et al., 2013*). In the Alps, chirotherian footprints were described from the upper Permian, Lower to Middle Triassic of the Dolomites, Piedmont and Ligurian Alps in Italy (*Avanzini & Mietto, 2008*; *Petti et al., 2013*; *Bernardi et al., 2015*; *Santi et al., 2015*), Aar Massif in eastern Switzerland (*Feldmann, Furrer & Glarus, 2009*; *Klein et al., 2016*) and the Aiguilles Rouges Massif (Western Alps), on the border between Switzerland and France (*Demathieu & Weidmann, 1982*; *Avanzini & Cavin, 2009*; *Cavin et al., 2012*; *Klein et al., 2016*).

We here describe and interpret a new archosaur track assemblage from the Gardetta Plateau (Western Alps, south-western Piedmont, Italy; Fig. 1) that was analyzed in two different field projects, during the summer 2009 and in the autumn 2017–2018.

Tracks are preserved on two distinct track surfaces, belonging to the same stratigraphic horizon. Some of them are badly preserved but distinct trackways, up to 3 m long, can be recognized together with other exceptionally preserved isolated tracks showing clear morphological details of the trackmaker's autopodium.

This discovery provides reliable evidence of the presence of archosauriforms in the Briançonnais domain during the Early Triassic, adding further support to the occurrence of terrestrial tetrapods at low latitudes soon after the PTME (*Bernardi et al., 2015*; *Bernardi, Petti & Benton, 2018*; *Romano et al., 2020*) and well-before a full land ecosystem recovery.

## MATERIALS AND METHODS

All the specimens were identified in the same outcrop, located about 1 km SE of the Gardetta Plateau, close to Pianezza creek (44°24′5.75″N; 7°1′45.29″E; Canosio Municipality, Cuneo Province, NW Italy; Fig. 1). Most of the footprints are preserved as negative epichnia (concave epirelief), and were discovered by EC and MP in summer 2008. A surface of about 10–15 m$^2$ was mapped for the first time in 2009 by HF and then in 2017 by FMP and HF. An exceptionally preserved trackway, consisting of three large pes and manus imprints, was then discovered during the 2017 and 2018 field seasons by EM and FMP, about 10 m higher up on the same outcropping horizon.

Three tetrapod trackways have been identified from the Gardetta outcrop: GT-1, GT-2 and GT-7. The GT-1 is composed by 4 consecutive footprints whereas GT-2 by 8 footprints. Both trackways were assigned to the ichnogenus *Chirotherium* and were left in situ in the outcrop. The trackway GT-7 is composed by three consecutive manus-pes sets and was assigned to *Isochirotherium gardettensis* ichnosp. nov. The Holotype GT-7-3 was left in situ but was also digitally modelled, printed and is now stored at the Museo di Geologia e Paleontologia dell' di Torino and digitally stored in MorphoSource (see below). On the same bedding surface where GT-1 and GT-2 trackways are preserved, few other isolated footprints were found (GT-3, GT-4, GT-5, GT-6). All of them were left in situ. GT-3 was assigned to *Isochirotherium* isp. GT-4, GT-5, GT-6 were not assigned to any existing

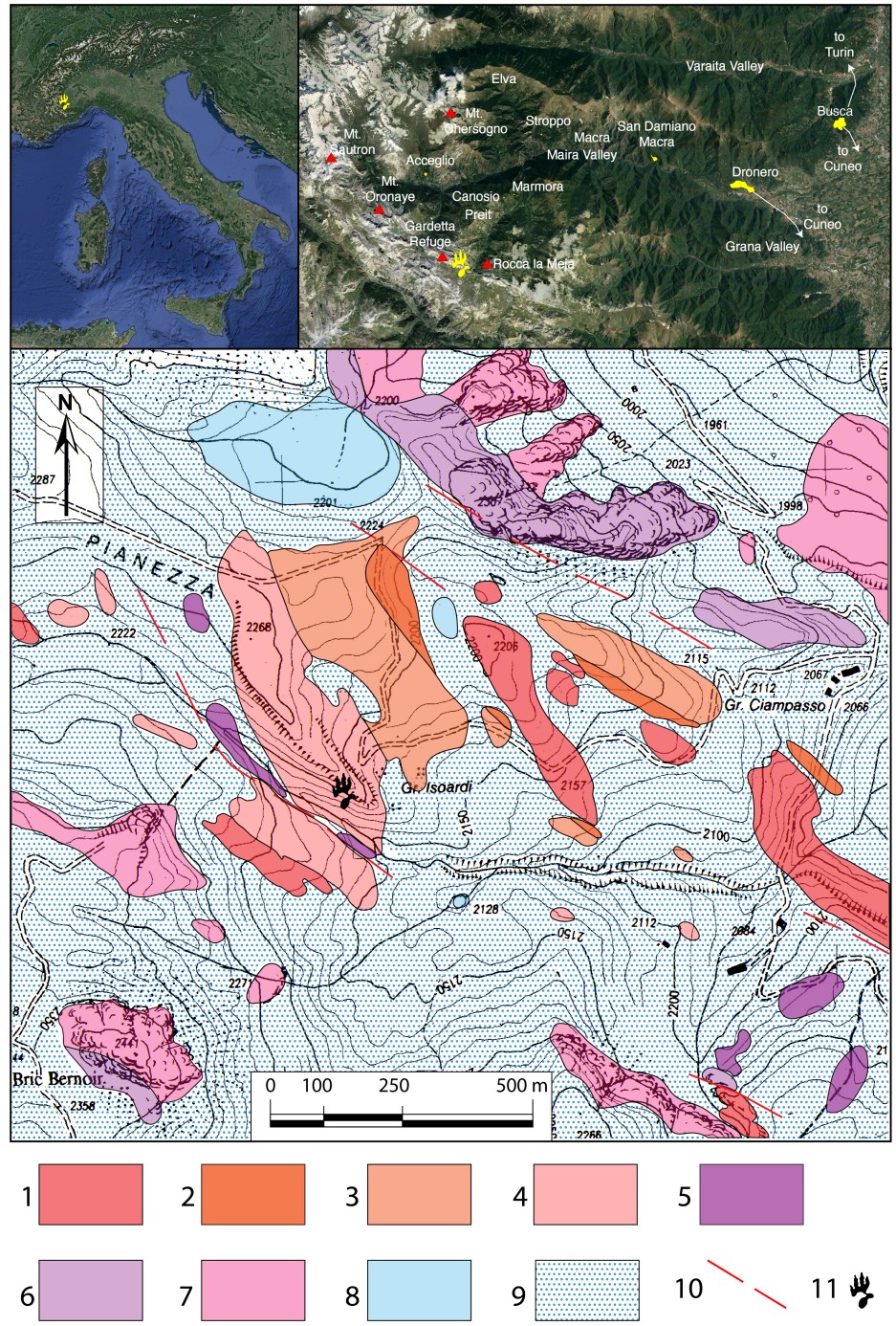

**Figure 1  Geologic map of the Pianezza area.** In the upper row the location of Maira Valley and Gardetta-Pianezza area. For the geologic map: 1 = volcanic complex and graphitic schist (upper Carboniferous—Permian); 2 = conglomerate, 3 = quartz-conglomerate, and 4 = quartz-arenite and quartz-siltite of the quartzitic complex (upper Permian—early Lower Triassic); 5 = lower carniole complex (late Lower Triassic); 6 = lower calcareous complex (lower Anisian—early upper Ladinian); 7 = upper dolomitic complex (upper Ladinian); 8 = lakes and peat bog; 9 = detritic cover and moraines; 10 = faults; 11 = location of the footprint site.

**Table 1  Ichnological material from Gardetta.** Specimen number, taxonomic affinities and repository for all the ichnological material from Gardetta considered in the present contribution.

| Number of specimen | Ichnotaxonomy | Repository |
|---|---|---|
| GT-1-1 | *Chirotherium* isp. | Left *in situ* in the outcrop |
| GT-1-2 | *Chirotherium* isp. | Left *in situ* in the outcrop |
| GT-1-3 | *Chirotherium* isp. | Left *in situ* in the outcrop |
| GT-1-4 | *Chirotherium* isp. | Left *in situ* in the outcrop |
| GT-2-1 | *Chirotherium* isp. | Left *in situ* in the outcrop |
| GT-2-2 | *Chirotherium* isp. | Left *in situ* in the outcrop |
| GT-2-3 | *Chirotherium* isp. | Left *in situ* in the outcrop |
| GT-2-4 | *Chirotherium* isp. | Left *in situ* in the outcrop |
| GT-2-5 | *Chirotherium* isp. | Left *in situ* in the outcrop |
| GT-2-6 | *Chirotherium* isp. | Left *in situ* in the outcrop |
| GT-2-7 | *Chirotherium* isp. | Left *in situ* in the outcrop |
| GT-2-8 | *Chirotherium* isp. | Left *in situ* in the outcrop |
| GT-3 | *Isochirotherium* isp. | Left *in situ* in the outcrop |
| GT-4 | unnamed footprint | Left *in situ* in the outcrop |
| GT-5 | unnamed footprint | Left *in situ* in the outcrop |
| GT-6 | unnamed footprint | Left *in situ* in the outcrop |
| GT-7-1 | *Isochirotherium gardettensis* | Left *in situ* in the outcrop |
| GT-7-2 | *Isochirotherium gardettensis* | Left *in situ* in the outcrop |
| GT-7-3 (Digital cast MGPT-PU1357853) | *Isochirotherium gardettensis* | Left *in situ* in the outcrop. 3D printing stored at the Museo di Geologia e Paleontologia dell'Università di Torino and digitally stored in MorphoSource |
| GD-E1 | unnamed footprint | Loose slab currently stored at the Museo di Geologia e Paleontologia dell'Università di Torino |
| GD-E2 | unnamed footprint | Loose slab currently stored at the Museo di Geologia e Paleontologia dell'Università di Torino |
| GD-E12 | unnamed footprint | Loose slab currently stored at the Museo di Geologia e Paleontologia dell'Università di Torino |

ichnotaxon because poorly preserved. GD-E1, GD-E2 and GDE-12 are specimens that were found as solitary footprints from loose slabs (erratic) and are currently stored at the Museo di Geologia e Paleontologia dell'Università di Torino. The specimen number, taxonomic affinities and repository for all the considered ichnological material from Gardetta is summarized in Table 1.

Tracks outlines were drawn on transparency acetate film and then digitized by a vector-based drawing software (Adobe Illustrator ©).

Close-range photogrammetry was used to document tracks and obtain three-dimensional model of the best-preserved ones (*Petti et al., 2008*; *Remondino et al., 2010*; *Mallison & Wings, 2014*). The data processing phase was performed using Agisoft PhotoScan® Professional software, following the procedure indicated by *Mallison & Wings (2014)*. In a second phase, the software Surfer®14 (*Golden Software, 2002*), was used to convert elevation points to contour lines and to produce color coded maps of the studied material. The obtained images are ideal for both precisely-measured standard

ichological parameters (*Haubold, 1971b*; *Leonardi, 1987*) and for recognizing anatomy related morphologies, therefore for the reconstruction of the trackmaker's autopodial osteology.

The search for a putative trackmaker was carried out employing three different and integrated methodological approaches: (i) Synapomorphy-based correlation (*Olsen, 1995*; *Carrano & Wilson, 2001*); (ii) Phenetic correlation (*Carrano & Wilson, 2001*) and (iii) Coincidence correlation (*Carrano & Wilson, 2001*). The synapomorphy-based method focuses on the identification of osteologic-derived character states in the footprints that result from the impression of synapomorphic characters in the trackmaker autopodia (see *Olsen, Smith & McDonald, 1998*; *Carrano & Wilson, 2001*; *Wilson, 2005*; *Romano, Citton & Nicosia, 2015*). The phenetic correlation is closely linked to ichnotaxonomy and derives from an accurate description of the footprint and the identification of the trackmaker through the recognition of an affinity between tracks and autopods osteology (*Carrano & Wilson, 2001*; *Wilson, 2005*). The coincidence correlation is usually adopted to refine trackmaker identification and is based on supplemental data including geological age, geographic provenance, local faunal composition and distributions, and abundances of skeletal taxa and ichnotaxa (*Carrano & Wilson, 2001*).

## Nomenclatural acts

The electronic version of this article in Portable Document Format (PDF) will represent a published work according to the International Commission on Zoological Nomenclature (ICZN), and hence the new names contained in the electronic version are effectively published under that Code from the electronic edition alone. This published work and the nomenclatural acts it contains have been registered in ZooBank, the online registration system for the ICZN. The ZooBank LSIDs (Life Science Identifiers) can be resolved and the associated information viewed through any standard web browser by appending the LSID to the prefix http://zoobank.org/. The LSID for this publication is: urn:lsid:zoobank.org:pub:654063D7-8AE9-4F4A-A6E9-E518F3D6B79E. The online version of this work is archived and available from the following digital repositories: PeerJ, PubMed Central and CLOCKSS.

## Geological framework

The Gardetta Plateau—Preit valley area is located in the southern part of the Western Alps (Fig. 1). It encompasses the Sautron, Rouchouze, Rocca Peroni tectonic units and the Gardetta deformation unit (*sensu D'Atri et al., 2016*) also known as "*bande siliceuse de la Gardetta*" (*Gidon, 1972*). These tectonic units pertain to the Briançonnais Domain (*Gidon, 1958a*; *Gidon, 1958b*; *Gidon, 1972*; *Schmid et al., 2004*; *Schmid et al., 2017*) and in particular to the External Briançonnais Domain which is characterized by very low grade to anchizone metamorphism (*D'Atri et al., 2016*).

The upper Permian-Mesozoic sedimentary succession varies considerably within the Briançonnais Domain *s.l.* (Briançonnais Domain *s.s.* and Ligurian Briançonnais, *Decarlis & Lualdi, 2009*; Fig. 2) due to the slightly different paleogeographic positions of these sectors (see *Decarlis et al., 2013* for a review). The outcropping lithostratigraphic units, even if they
can be correlated across the distinct domains, display different thickness, vertical/lateral relationships and hiatuses. These differences led authors to adopt a multitude of official and unofficial names for the lithostratigraphic units. Despite these minor differences, the late Permian–Early Triassic sedimentation in the whole Briançonnais domain *s.l.* testifies to the evolution of a continental margin affected by extensional tectonics. The Briançonnais domain was positioned north of the westernmost sector of the Palaeotethys, in the western continental termination of the Meliata oceanic back-arc basin (*Ziegler & Stampfli, 2001*; *Decarlis et al., 2013*). Adopting the paleolatitude calculator developed by *Van Hinsbergen et al. (2015)* (model version 2.1) and using the Global Apparent Polar Wander Path of *Torsvik et al. (2012)* as the paleomagnetic reference frame, the Early Triassic (250 Ma) palaeolatitude estimate for the Southern Briançonnais Domain is 11.8 N.

In the study area the volcano-sedimentary succession starts with upper Carboniferous—Permian volcanic rocks (andesitic lavas followed by rhyolites and rhyolitic ignimbrites) unconformably overlain by upper Permian-Lower Triassic siliciclastic continental-to-transitional deposits (the so called "*semelle silicieuse*" of French authors). In particular these deposits are characterized by basal coarse grained conglomerates and quartz-conglomerates, named locally "*Verrucano Brianzonese*" (*Carraro et al., 1970*; *Cassinis, Perotti & Santi, 2018*), that evolve upward into quartz-arenites and quartz-siltites of the "*Werfenian quartzites*" (Fig. 2; *Gidon, 1958b*; *Malaroda, 1970*; *Megard-Galli & Baud, 1977*; *Costamagna, Barca & Nervo, 2002*; *Costamagna, 2013*). The siliciclastic sequence indicates deposition in an alluvial environment characterized by sandy braided fluvial system fed by the residual Variscan relief (*Costamagna, 2013*). In the southernmost part of the Briançonnais domain (External Ligurian Briançonnais Domain, *Vanossi, 1974*; *Vanossi, 1991*; *Bertok et al., 2012*) these latter lithostratigraphic units are known as "*Scytian quartzites*" or "*Ponte di Nava Quarzites*" (Fig. 2; *Decarlis et al., 2013*; *Decarlis et al., 2015*). Similar to the siliciclastic sequence of the Briançonnais Domain *s.s.*, the "*Ponte di Nava Quarzites*" originated from the dismantling and reworking of the Paleozoic igneous and metamorphic basement.

The quartz-arenites can be topped either by greenish pelites (known as "*Case Valmarenca Pelites*" in the Ligurian Briançonnais, *Vanossi, 1974*; *Vanossi, 1991*), that have been interpreted as mudflat deposits, or by a thin and discontinuous interval of cavernous dolostones called "*Cargneules Inférieures*" representing the sedimentation in an arid environment as an evaporitic sabkha (Fig. 2). According to *Lualdi & Seno (1984)*, in the Ligurian Briançonnais Zone the "*Case Valmarenca Pelites*" could be laterally equivalent to the "*Cargneules inférieures*".

The continental succession and/or the evaporitic deposits are followed by Middle Triassic shallow water carbonates of the "*couverture carbonatée*" (*Gidon, 1958b*; *Megard-Galli & Baud, 1977*; *Costamagna, Barca & Nervo, 2002*) comprising a lower calcareous complex (Costa Losera Formation, *Lualdi & Bianchi, 1990*, corresponding to the e St. Triphon Formation of the classic Briançonnais Domain) and an upper dolomitic complex (San Pietro dei Monti Formation, *Vanossi, 1969*). These carbonate deposits testify the sedimentation in a subsiding carbonate ramp.

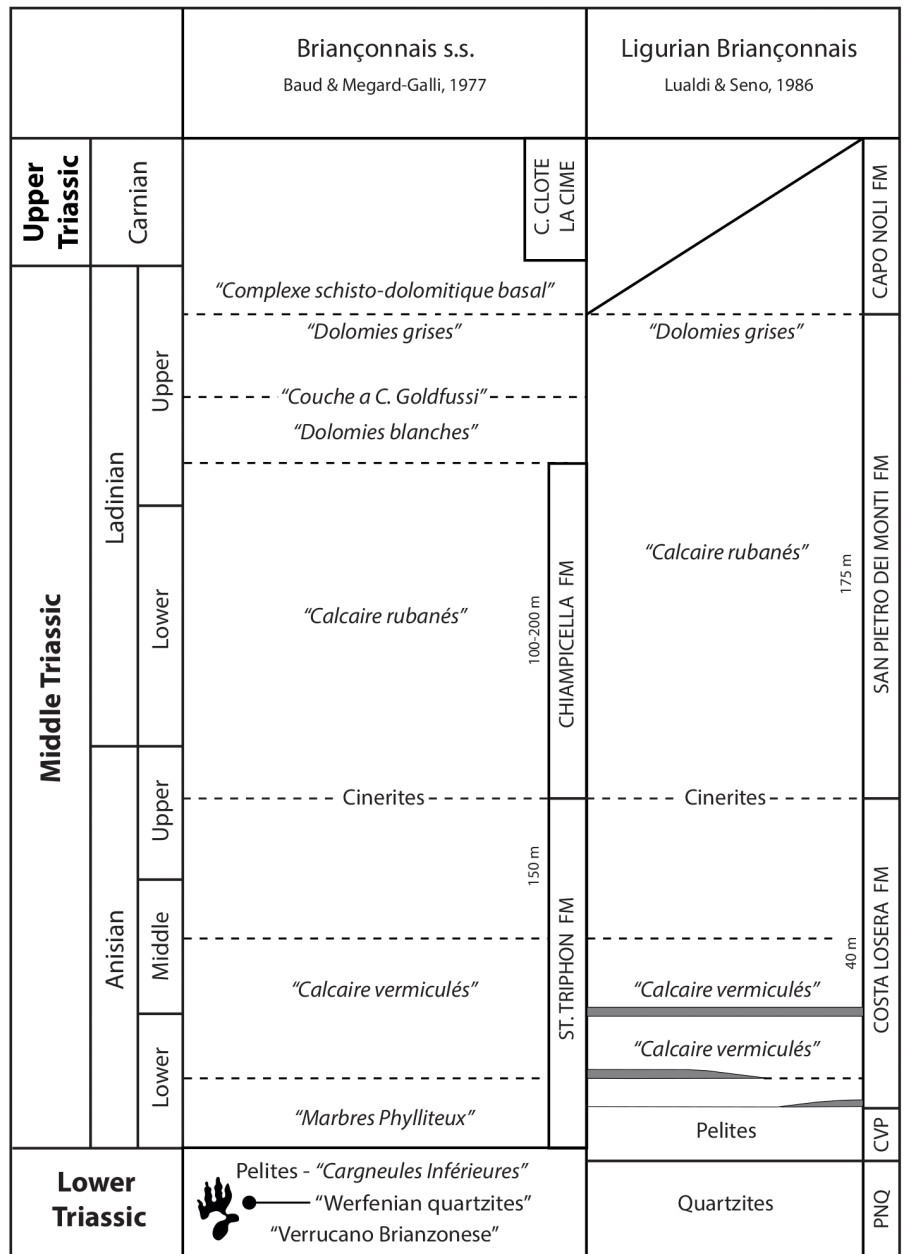

**Figure 2  Correlation scheme among the Briançonnais s.s., the Ligurian Briançonnais, from *Decarlis & Lualdi, 2009* redrawn and modified.** PNQ: ''Ponte di Nava Quartzites'', CVP: Case Val Marenca Pelites. The footprint silhouette marks the position of the track-bearing horizon.

The lower calcareous complex (Fig. 2) begins with a characteristic facies named "*Marbres Phylliteux*" by French authors made of greyish and brownish fine-grained limestones, (lower to upper Anisian) with sericite, muscovite, chlorite laminated levels. Bedding can be locally masked by pervasive and intense bioturbation (''*Calcaires Vermiculés*'' facies) assigned to the ichnogenus *Rhizocorallium*. The basal complex ends with varicolored

pelites, interpreted as cinerites (upper Anisian in age) by *Caby & Galli (1964)*, recognizable throughout the whole Briançonnais Domain.

The upper dolomitic complex (Fig. 2) is comprised of massive to well-bedded dolostones followed by cyclically arranged carbonates ("*Calcaires rubanés*"—upper Anisian—upper Ladinian; *Gidon, 1958b*; *Megard-Galli & Baud, 1977*; *Costamagna, Barca & Nervo, 2002*; *Decarlis & Lualdi, 2009*) characterized by subtidal crinoidal wackestones, intertidal oolitic limestones and supratidal dolomitic mudstones capped by reddish paleosols, that testify shallowing-upward cycles. The dolomitic succession includes dark limestones, dark fossiliferous and/or oolithic dolostones, meter-thick autoclastic breccias and gypsum–anhydrite pseudomorphs witnesses of major emersion events. These lithofacies, dated to the uppermost Ladinian, are known in the different Briançonnais domains as "*Dolomies blanches*" or "*Dolomies grises*" or "*Couches a C. goldfussi*" or "*Complexe schisto-dolomitique basal*".

## The Pianezza stratigraphic succession

In the framework of the above mentioned stratigraphic setting the footprint-bearing level is located in the Pianezza area along the track connecting Colle del Preit (2100 m a.s.l.) to Grange Isoardi (Pianezza area, 2,275 m a.s.l.) (Fig. 2). The outcrop is located along the SW flank of a narrow antiformal anticline belonging to the Sautron Tectonic Unit which overthrusts the Rouchouze Tectonic Unit. Here the volcano-stratigraphic succession begins by meta-andesites and andesitic schists pertaining to the upper Carboniferous-Permian volcanic complex. The sedimentary succession continues upward with a thin and discontinuous (up to 1 meter) level of graphitic schists, deriving from the weathering of the volcanic basement (*Lorenzoni & Zanettin, 1958*) and is then followed by up to 100 m of quartz-conglomerates ("*Verrucano Brianzonese*") and by fine to very fine quartz-arenite and quartz-siltite with ripple marks and cross bedding ("*Werfenian quartzites*"). The track-bearing horizon occurs at the top of the latter clastic interval. The succession continues upward with 15 m of gypsum/anhydrite deposits of the lower cargneule. In the Pianezza area the Middle Triassic "*couverture carbonatée*" is only represented in the north-eastern flank of Sautron Unit anticline.

## Chronostratigraphic framework of the study area

The sedimentary rock belonging to the quartz-rich clastic succession does not allow precise dating because of the lack of biostratigraphic markers as commonly happen for these kind of deposits. They are here referred to the upper Permian-Lower Triassic at the base of their stratigraphic position in the Sautron Unit, similar to that of the very comparable quartz-conglomerate and quartzarenite rocks occurring not only in the Briançonnais Domain, but also in the Southern Alps, Sardinia and Provence. For this reason, in order to constrain the age of the track-bearing horizon, some considerations are required: (i) the coarse quartz-conglomerates ("*Verrucano Brianzonese*") are commonly referred to the late Permian-earliest Triassic (*Gidon, 1958b*; *Carraro et al., 1970*; *Megard-Galli & Baud, 1977*; *Decarlis & Lualdi, 2009*); (ii) the Lower Triassic age can be hypothesized considering the occurrence of *Estheria minuta* and *Myacites fassaensis* within the "*Ponte di*

*Nava Quarzites*" (*Decarlis & Lualdi, 2009*); (iii) the "*lower cargneule*" unit and its lateral equivalent "*Case Val Marenca Pelites*" are generally attributed to the late Early Triassic (*Gidon, 1958b*; *Carraro et al., 1970*; *Megard-Galli & Baud, 1977*; *Decarlis & Lualdi, 2009*); (iv) the lower parts of "*Marbres Phylliteux*" are considered early Anisian in age, on the basis of the occurrence of *Rhizocorallium*, that is regarded as an early Anisian marker all over the Tethyan realm (*Baud, 1976*); (v) an early Anisian age for the base of the lower calcareous complex ("*Marbres Phylliteux*" and Costa Losera Formation) is also suggested by the occurrence of Dasycladacean algae and crinoidal remains (*Dadocrinus* sp.; *Carraro et al., 1970*); (vi) in the northern Briançonnais of southwestern Switzerland a find of the ammonoid *Beyrichites cadoricus* in the upper part of the St-Triphon Formation indicate a middle Anisian age (*Baud et al., 2016*).

Additionally, it is worth mentioning that both in the Geological Map of the Argentera Massif (*Malaroda, 1970*; *Carraro et al., 1970*) and in the Geological Map of France at the scale 1: 50.000 (Sheet 896, Larche; *Gidon, 1978*) the studied outcrop was attributed to Lower Triassic. All the above reported data thus point to a probable attribution of the trampled horizon to the late Early Triassic.

## Systematic ichnology

Most footprints are preserved as natural molds (concave epirelief) on top of a 3–4 cm thick bed of fine sandstone. The tracks are shallow, less than 2 cm deep, but most of them are cut by small-scale tectonic cracks/fissures and strongly weathered. Two possible trackways with lengths of 4–5 m were identified on a track surface. Only one isolated track was visible on the underlying sandstone bed, also preserved as concave epirelief, possibly undertracks of the upper level. Three solitary small footprints (GD-E1, GD-E2, GD-12), preserved as convex epirelief of the directly overlying sandstone bed, were collected from loose slabs and are currently stored at the Museo di Geologia e Paleontologia dell'Università di Torino (Turin, Italy). The upper surface of this 1–2 cm thick sandstone bed is marked by symmetric wave ripples, exposed on a spectacular bedding plane (Fig. 3).

An exceptionally preserved trackway, comprised of three consecutive manus-pes sets was found on another surface, belonging to the same stratigraphic horizon, upstream of the previously described ones (Fig. 4). The general features of the herein studied ichnoassemblage are typical for chirotherian tracks (*Haubold & Klein, 2002*).

**Ichnogenus *Chirotherium* Kaup, 1835**
**Type ichnospecies:** *Chirotherium barthii Kaup, 1835*
*Chirotherium* isp.
(Figs. 3 and 5)

**Referred specimens**: two trackways preserved as concave epirelief (GT-1 and GT-2). GT-1 consists of four clear and two weakly impressed imprints, arranged in a 2.10 m-long trackway in the lower part of the outcrop, just 2 m above the creek level (Fig. 3). Its direction on the steep bedding plane points upwards to southeast. Trackway GT-2 is 2.40-m-long,

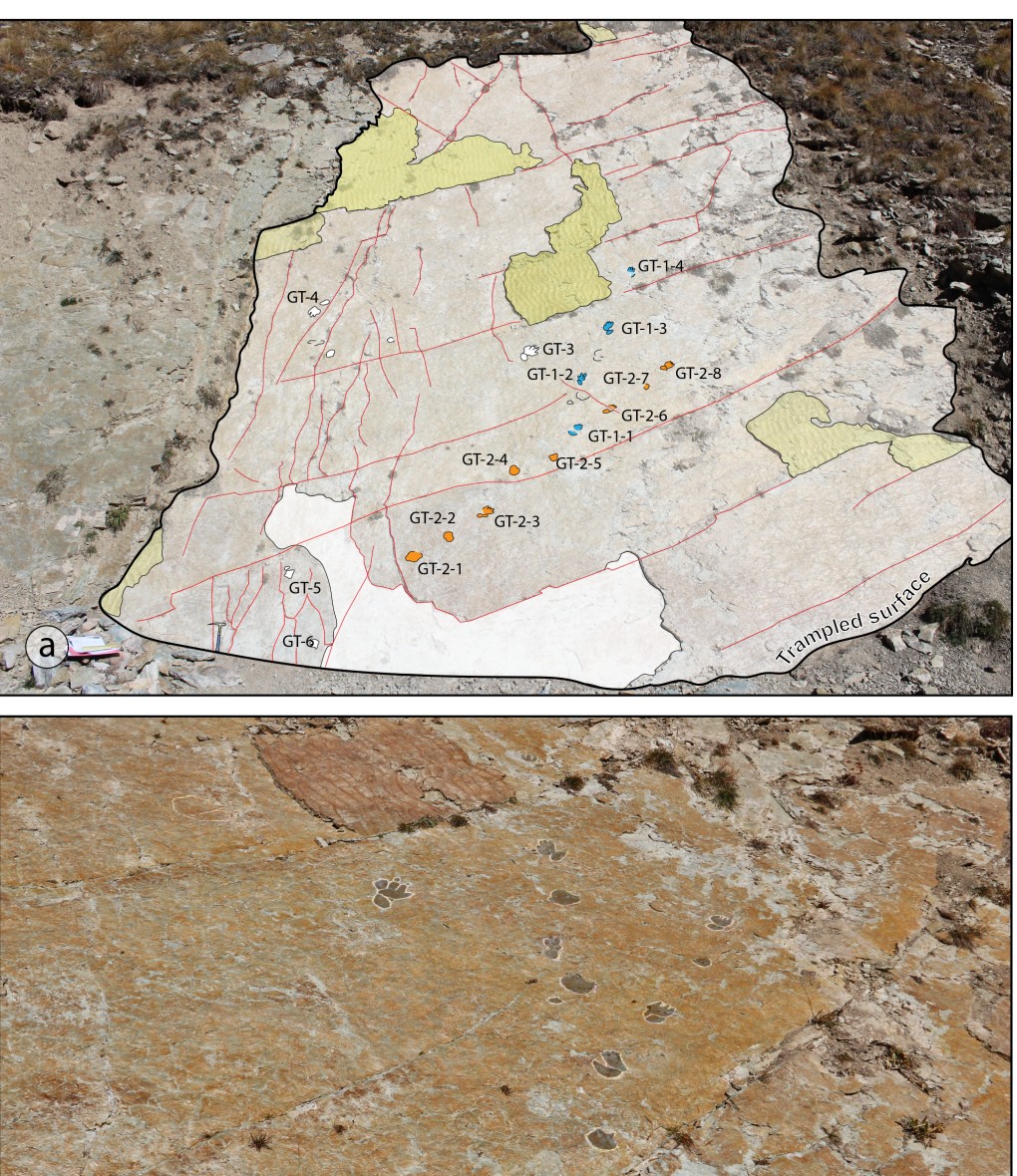

**Figure 3** (A) Panoramic view of the track surface with the line-drawing of the chirotherian trackways. In pale yellow the above-lying bed characterized by symmetric wave ripples; (B) detailed view of the GT-1 and GT-2 trackways, highlighted with the black colour.

is preserved in the lower part of the same bedding plane, about 2 m above the creek level, NW-SE oriented.

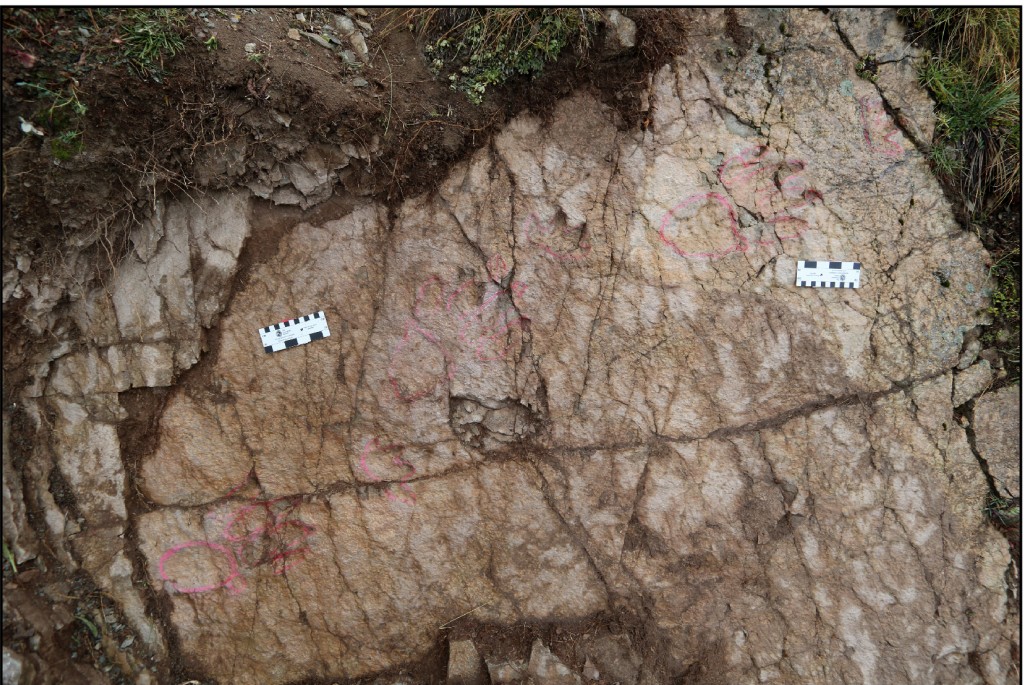

**Figure 4** **The GT-7 trackway of *Isochirotherium gardettensis* ichnosp. nov.** *Isochirotherium gardettensis* ichnosp. nov. The GT-7 trackway, made of three consecutive manus-pes couples, is here highlighted by the red chalk and preserved in the upper track-bearing surface. Scale bar: 13 cm.

**Description:** pentadactyl and semi-digitigrade pes imprint. Pes is longer than wide, (Foot Length [FL] = 13 to 16 cm; Foot Width [FW] = 8–10 cm; FL/FW = 1.6 to 2.0) with digit group II–IV roughly asymmetrical. Pedal digit impressions gradually increase from I to IV, with II sub-equal or shorter than digit IV; digit III is the longest. In the best-preserved track (GT1-3; Figs. 3 and 5), digit I impression is pointed and placed posteriorly with respect to digit group II-IV. Digit V impression is oval and tapers distally; it is positioned posteriorly and laterally to digit I-IV imprints and directed antero-laterally. No digital pad impressions can be observed on digit II-IV. Digit V impression shows a large rounded pad impression and a possible sub-triangular shaped claw mark. Manus tracks are absent or faintly preserved as small semi-circular imprints, placed in front of the pedal footprints. An isolated tetradactyl imprints, measuring 4.5 cm in length and 7 cm in width, and another isolated circular pentadactyl imprint 5.5 cm long are interpreted as possible manus imprints.

In the trackway the oblique pace varies between 26 and 41 cm, with a mean value of 36 cm. The pes pace angulation varies between 145° and 165°, with a mean value of 157°.

**Discussion:** the ichnogenus *Chirotherium* with its holotype *Chirotherium barthii*, was described by *Kaup (1835)* on trackways from the ''*Thüringischer Chirotheriensandstein*'' (Lower-Middle Triassic) of the Thuringia region (Germany). The here described material, even if not perfectly preserved, retains some diagnostic features of the ichnogenus *Chirotherium*, such as the oval morphology and the position of digit V imprint (slightly

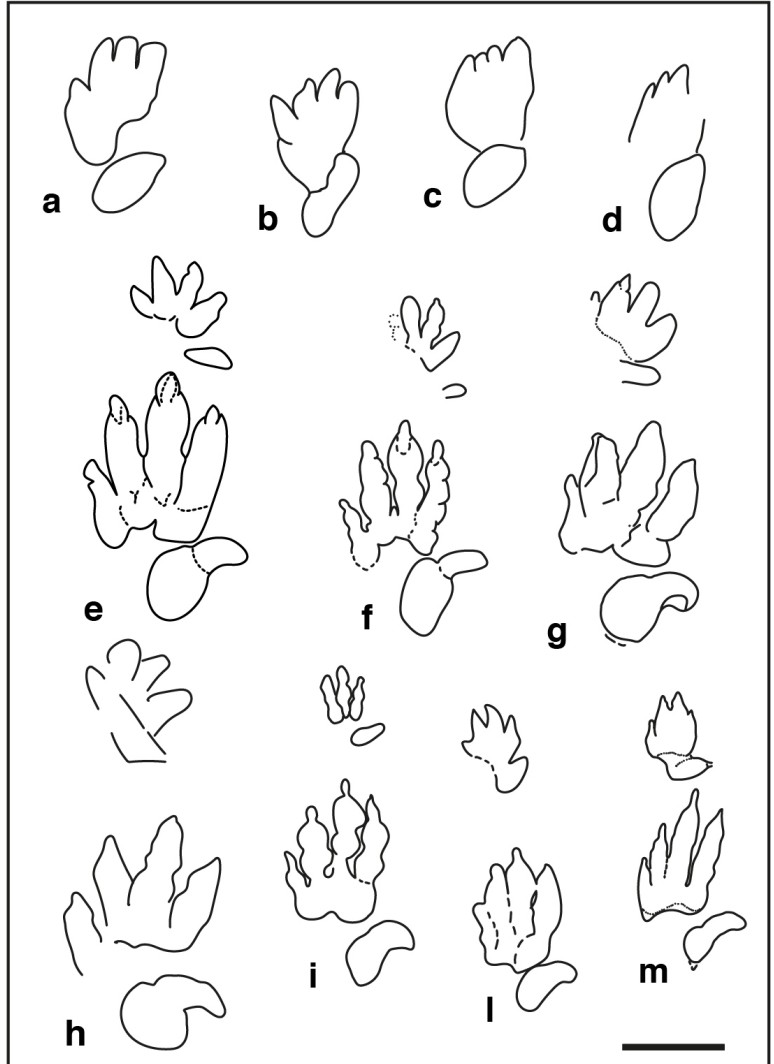

**Figure 5** Pentadactyl tracks from the Lower and Middle Triassic, assigned to the ichnogenus *Chirotherium* and their comparison with the studied tracks of the Gardetta ichnosite: (A) GT-1-3; (B) GT-2-3; (C) GT-2-8; (D) GT-2-6; (E), (F) *Chirotherium barthii* pes manus sets from type surface of the "Thüringischer Chirotheriensandstein", Hildburghausen, Germany; (G) *Chirotherium barthii* pes manus set from the Holbrook Member of the Moenkopi Formation (Middle Triassic), southwest of Cameron, northern Arizona; (H) *Chirotherium vorbachi* pes manus set from the Lower Triassic of Aura an der Saale, Germany; (I), *Chirotherium sickleri* "Thüringischer Chirotheriensandstein", Germany; (L) , (M) *Chirotherium sickleri* pes manus sets from the Wupatki Member of the Moenkopi Formation (Lower Triassic), Meteor Crater, Arizona. Scale bar 10 cm.

behind digit group II–IV), and the relative digit length of group II–IV, with digit IV longer or sub-equal to digit II impression. Pes pace angulation is also similar to the values to date reported for the ichnogenus (160° − 170°). *Chirotherium barthii* (Figs. 5E and 5F) shows clear circular pads on digit group II-IV and digit impressions are broader than in the studied specimens. In *C. barthii*, as well as in *C. rex*, *C. moquinense* and *C. vorbachi* (Fig.

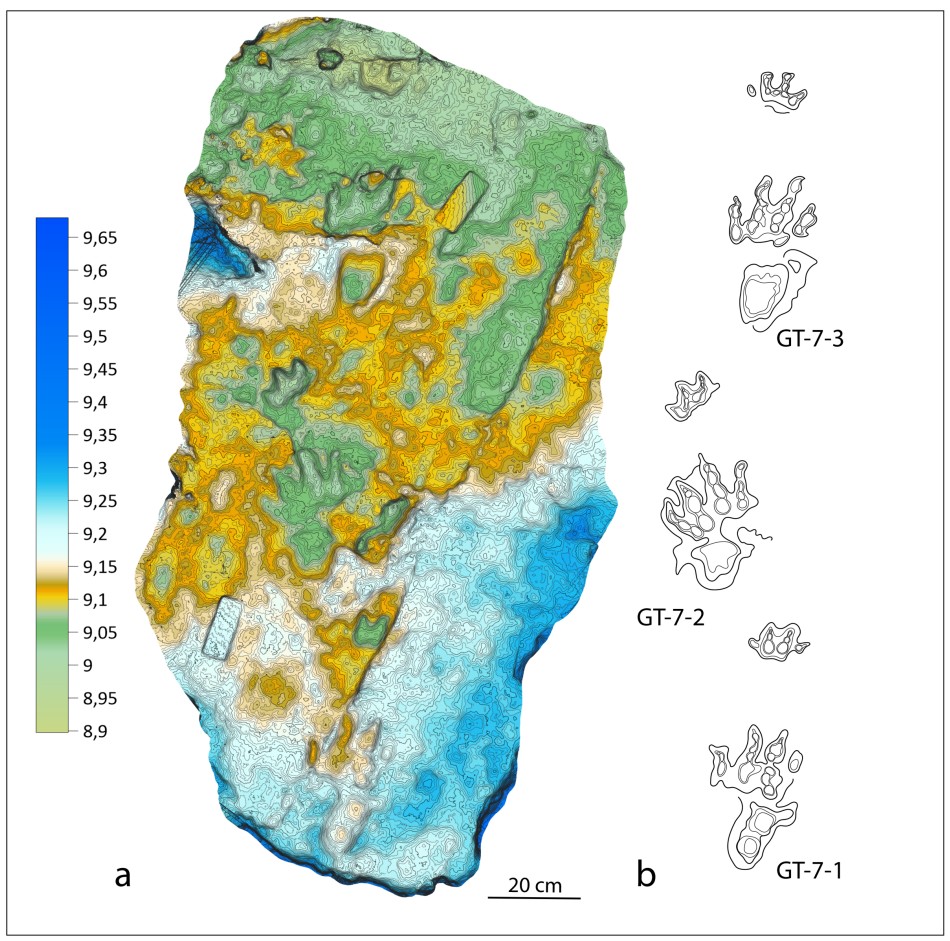

**Figure 6** **Colour-coded, contour line and interpretative drawing of the GT-7 trackway.** (A) *Isochirotherium gardettensis* ichnosp. nov. Colour-coded and contour line image of the GT-7 trackway; (B) Interpretative drawing of the GT-7 trackway.

5H), digits I–IV imprints are splayed whereas in the GT-1 and GT-2 trackways, pedal digits outlines are closely arranged with only digit I impression medially spread. Digits II–IV imprints seems to be almost parallel to each other and the digit pattern resemble that of the ichnospecies *C. sickleri* (*Kaup, 1835*) (Figs. 5I, 5L and 5M) with digit I forming a narrow group with digits II, III and IV. Nevertheless, digit IV impression, though slightly shorter than III, is not much longer than II as observed in most of the specimens assigned to *C. sickleri*. Unfortunately, the bad preservation of pes imprints in GT-1 and GT-2 trackways precludes any accurate ichnospecific assignment.

Ichnogenus *Isochirotherium Haubold, 1971a* (Figs. 4, 6–8)
**Type ichnospescies**: *Isochirotherium soergeli* (*Haubold, 1967*).
*Isochirotherium gardettensis* ichnosp. nov.

**Derivatio nominis:** from the Gardetta plateau, type locality of the ichnospecies.

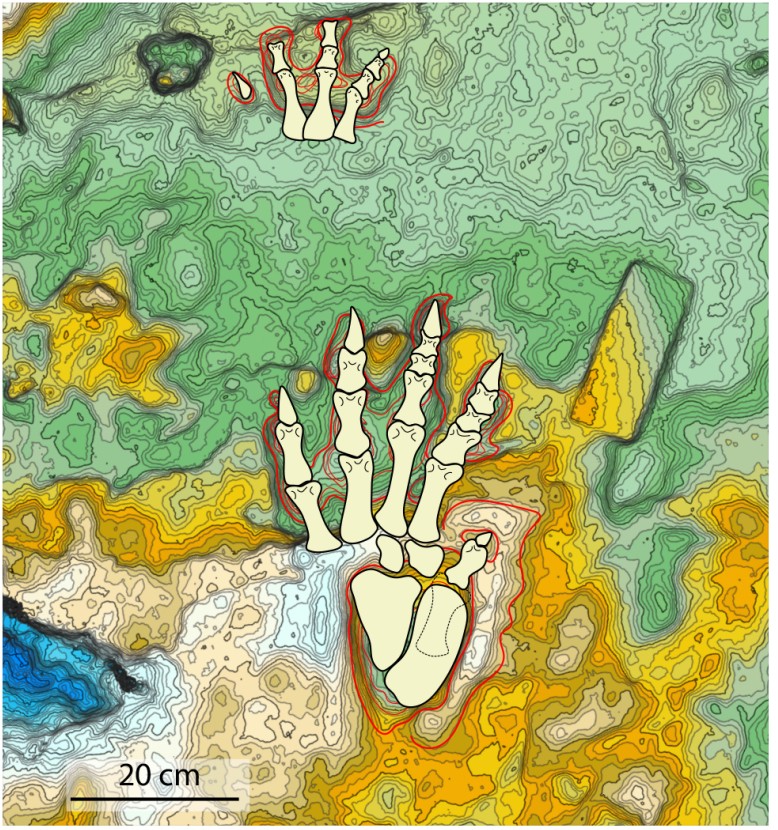

**Figure 7** **Reconstruction of the trackmaker's fore- and hind limbs, based on the 3D model and its interpretative drawing.** Dashed lines define the metatarsal of digit V held lifted off the ground during locomotion.

**Type-level:** "*Werfenian quartzites*", Lower Triassic.

**Holotype:** GT-7-3 manus-pes couple left in situ. A cast (MGPT-PU135785, Museo di Geologia e Paleontologia dell'Università di Torino, Italy) was printed after the 3D-modelling of the GT-7-3 manus-pes couple. The 3D model of the holotype is digitally stored in MorphoSource at the permanent link https://www.morphosource.org/Detail/SpecimenDetail/Show/specimen_id/45431

**Referred specimens:** a trackway made of three exceptionally-preserved (value 3 of the numerical scale proposed by *Belvedere & Farlow, 2016*) and consecutive manus-pes couples (GT-7; Fig. 4) not exceeding 2.20 m across. Another possible isolated track (GT-3) partially preserved in the lower track surface.

**Diagnosis**: chirotherian track with pentadactyl pes impression and small and tetradactyl manus imprint; pes digit IV impression noticeably shorter than II; pes digit group I-IV imprint slightly longer than wide, pes digit V impression with large ovoid pad and a reduced phalangeal portion. Manus is digitigrade with no impression of digit proximal sole-pad (i.e., ulnare-radiale fleshy pad) and digit V.
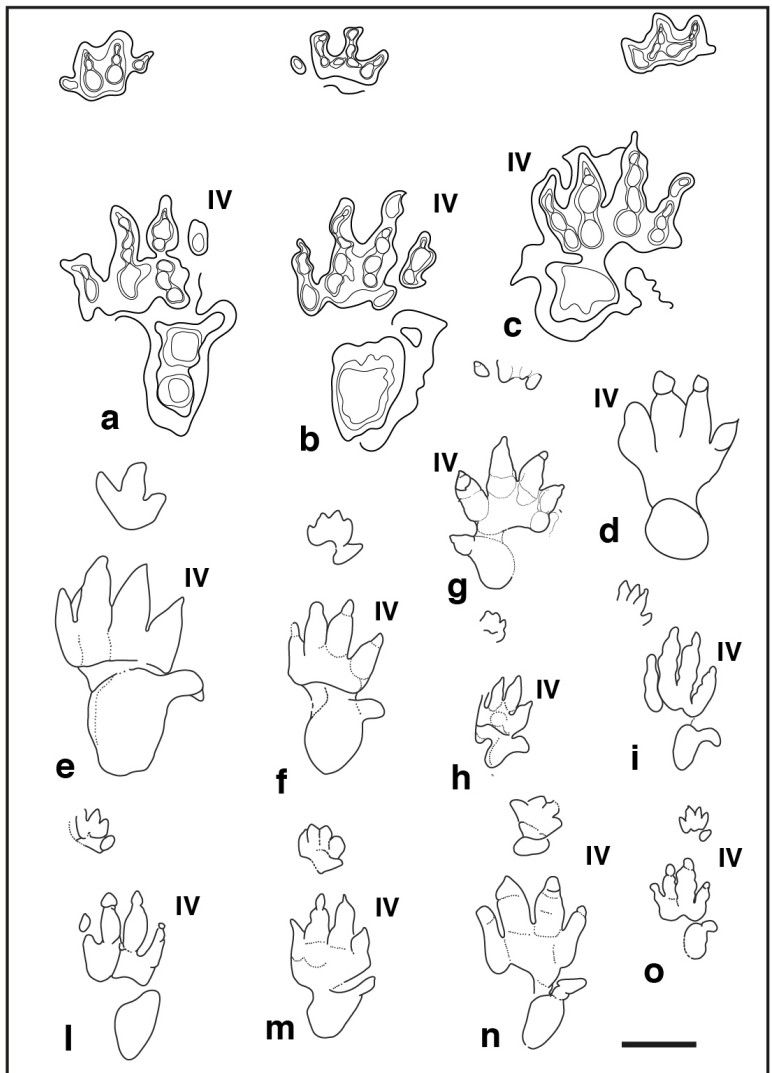

**Figure 8** **Pentadactyl tracks from the Lower and Middle Triassic, assigned to the ichnogenus *Isochirotherium* and their comparison with the studied tracks of the Gardetta ichnosite:** (A), (B), (C), pes manus sets of the GT-7 trackway; (D) GT-3 isolated pes imprints of the lower track surface; (E) *Isochirotherium herculis* pes manus set from the "Thüringischer Chirotheriensandstein" (Lower Triassic), Germany; (F) *Isochirotherium marshalli* pes manus set from the Holbrook Member of the Moenkopi Formation (Middle Triassic), Penzance, Northern Arizona; (G) *Isochirotherium infernense* manus pes set from the Middle Triassic (late Anisian) of Adige Valley, Bolzano, Italy; (H) *Isochirotherium coltoni* pes manus set from the Wupatki Member of the Moenkopi Formation (Lower Triassic), Meteor Crater, Arizona; (I) *Isochirotherium lomasi* pes manus set from the Middle Triassic (Anisian) of Cheshire, Great Britain; (L) *Isochirotherium coureli* pes manus set from the Middle Triassic (Anisian-Ladinian) of the Massif Central, France; (M) *Isochirotherium hessbergense* pes manus set from the "Thüringischer Chirotheriensandstein" (Lower Triassic), Germany; (N) *Isochirotherium demathieui* pes manus set from the Middle Triassic of Mont d'Or Lyonnais, France; (O) *Isochirotherium soergeli* pes manus set from the "Thüringischer Chirotheriensandstein" (Lower Triassic), Germany. Scale bar 10 cm.

Digit II and III impressions fairly parallel both in manus and pes. Manus consistently smaller than the pes and placed medially in front of the pes; manus digit III impression

parallel to digit III trace of the pes. Cross axis equal to 90°. Trackway very narrow, pace angulation near 165°, and ratio of stride to pes length is 4.3.

**Description:** pentadactyl and semi-plantigrade pes imprint, longer than wide (FL = 33.4 cm; FW = 19.2 cm; FL/FW = 1.74). Digit III impression is the longest. It is slightly longer than II, whereas digit IV imprint is shorter than II. Digit I impression is the shortest and is thinner than those of digit group II–IV. The total divarication I–IV is 22°; the angle between digit I and II is 8° and is equal to that between the impressions of II and III but larger than II–IV (6°). Cross axis is nearly equal to 90°. Digit impressions are robust and pointed showing large sub-triangular claw marks. Three to four phalangeal pad impressions are present on each digit of group I–IV. The impression of metatarsal-phalangeal portion is proximally arched and could be separated from digit V by a gap, or joined with it through a convex area, running from the basalmost portion of digit I to the medial digit V. Digit V impression shows a large oval impression joined to a rounded phalangeal-ungual portion, laterally spread out. In GT-7-2 and GT-7-3, pes digit V impression has a sub-triangular shape with a wider inner margin, almost aligned with the medial margin of digit I. Length of pes digits impressions are: (I) 118 mm; (II) 173 mm; (III) 186 mm; (IV) 136 mm; (V) 167 mm.

The manus print is small, tetradactyl and digitigrade, wider than long (FL = 8.04 cm; FW = 13 cm; FL/FW = 0.62) and is placed in front of the pes print. Digit impressions are short and pointed. Digits II and III have nearly equal length and are longer than digits I and IV; the latter is moderately spread outward. One pad impression is visible on digits I and IV whereas three pads characterize digits II and III. Digit IV impression is possibly the shortest. Length of manus digits are: (I) 49 mm; (II) 74 mm; (III) 68 mm; and (IV) 43 mm.

The trackway, made by three consecutive manus-pes sets, shows a clear narrow gait (pace angulation 164°). Oblique pace is 59 cm, whereas double pace is 119 cm across. Manus-pes couples turned slightly outward with respect to the midline (from 10° to 15° on average).

**Discussion**: the ichnogenus *Isochirotherium* was erected by *Haubold (1971a)*; its type ichnospecies *I. soergeli* (*Haubold, 1967*) comes, as for *Chirotherium barthii*, from the "*Thüringischer Chirotheriensandstein*" (Lower-Middle Triassic) of the Thuringia region (Germany). The ichnogenus is reported also from the Middle Triassic of Great Britain (*Treasise & Sarjeant, 1997*; *King et al., 2005*), from the Lower–Middle Triassic of North American Southwest (*Peabody, 1948*; *Klein & Lucas, 2010a*; *Klein & Lucas, 2010b*), the Aiguilles Rouges Massif (Western Alps) on the border between Switzerland and France (*Avanzini & Cavin, 2009*; *Klein et al., 2016*) and from the Middle Triassic of North-East Italy (*Avanzini & Leonardi, 2002*).

The main diagnostic features of this ichnogenus, recognized in our specimens are: (i) the relative digit length, with digit II longer than IV and shorter than III; (ii) a marked heteropody; (iii) the pes pace angulation around 165°; (iv) the weakly impressed distal portion of digit V and (v) pes-manus couples outward rotation of about 15°. However, the studied trackway shows clear difference to most of the ichnospecies known to date. For example, the type ichnospecies *I. soergeli* (*Haubold, 1967*) (Fig. 8O), has smaller absolute dimensions (even if size is not necessarily diagnostic), thinner pes digit marks and, most

importantly, display five clear digit impressions in the manus contrary to GT-7, where only tetradactyl manus were observed.

*Isochirotherium hessbergense* (*Haubold, 1971a*) (Fig. 8M) has also a pentadactyl manus placed more closely to pes digit tips and is clearly different from the material described herein for its digit group I–IV longer than wider, for the relative pes digit length, (notably digit I is longer than IV) and proportionally stouter digit impressions.

*Isochirotherium demathieui* (*Haubold, 1971a*) (Fig. 8N) can be excluded for its pentadactyl manus and for the shorter distance between manus and pes. Additionally, digits III and IV pes impressions are stouter than in *I. gardettensis* and the medial embayment is more pronounced; sole-pad pes impression in *I. demathieui* is consistently smaller.

*Isochirotherium coltoni* (*Peabody, 1948*) (Fig. 8H) and *I. lomasi* (*Baird, 1954*) (Fig. 8I) retain much slenderer digit impressions, especially in the pes imprint and most notably have manus tracks more internally placed and placed closely to pes digit tips than in the studied material. The proximal sole-pad pes impression is consistently smaller than *I. gardettensis*, especially in *I. coltoni*. *I. lomasi* differs from all other ichnospecies in having all digit phalangeal pes impression quite well-separated for their entire length. In addition, in *I. lomasi* pes digit group I–IV is detached from digit V.

*I. herculis* (*Egerton, 1839*) (Fig. 8E) can also be excluded for (i) the tridactyl manus; (ii) the digit group I-IV slightly wider than longer and (iii) the manus imprint position, very close to that of the pes and placed in front of pes digit tips II–III whereas in *I. gardettensis* is projected medially in front of digit II tip In *I. herculis* pes digit traces are stouter than in the studied specimens and the sole-pad is consistently broader with a fairly absent medial embayment.

*Isochirotherium marshalli* (*Peabody, 1948*) (Fig. 8F) shows similar features such as: (i) the pes digit relative length; (ii) the interdigital angles values; (iii) the digit group I-IV as longer as wider; (iv) the arched metatarsal-phalangeal portion; (v) the configuration of digit V whose phalangeal portion is significantly smaller than the ovoidal and possibly tarsal-metatarsal pad. Furthermore, in *I. marshalli* the proximal pes sole-pad impression is more centrally located behind digit II and III whereas in *I. gardettensis* lies more externally behind digit group III–V.

*Isochirotherium infernense* (*Avanzini & Leonardi, 2002*) from the Illyrian (late Anisian, Middle Triassic; Fig. 8G) of the Adige Valley (Bolzano, NE Italy) closely resembles the Gardetta specimens for: (i) the arched metatarsal-phalangeal portion; (ii) the position of the base of pes digit V, placed along the axis of digit III; (iii) pes digit relative length; (iv) cross axis equal to 90° (v) pes angulation of about 160°; (vi) positive rotation of manus-pes couples respect to the midline (10° − 15°). However, pes digits are stouter and the manus is described as pentadactyl (even if in the outline drawing only four digits are clearly appreciable) and more importantly interdigital angles are consistently wider, especially between digits II and III that are roughly parallel in *I. gardettensis*. Manus tracks in *I. infernense* is located frontally to digit II and III rather than medially as in *I. gardettensis*. Sole-pad pes impression is consistently smaller than in *I. gardettensis*.

The tracks referred to *Isochirotherium delicatum* (*Courel & Demathieu, 1976*) and found in the Anisian-Ladinian deposits of Argentière (Ardèche, France; *Courel & Demathieu,*
*1976*; *Courel, Demathieu & Gall, 1979*; *Demathieu, 1984*; *Gand, 1978*) and Gampempass (Southern Alps, Italy; *Avanzini & Lockley, 2002*) show less-thick digit impressions and a markedly reduced digits IV and V; the latter is also much more backward positioned if compared with the studied specimens. Overall pes impression in *I. delicatum* is consistently longer than wider with a laterally-compressed general appearance.

We therefore erect the new ichnospecies *Isochirotherium gardettensis* to describe a new and exceptionally-preserved *Isochirotherium* trackway that differs from all the other ichnospecies for all the features listed above.

## Searching for a putative trackmaker

Grounding on previous studies and new observations, *Bernardi et al. (2015)* showed that chirotherian footprints, such as *Protochirotherium*, *Chirotherium*, *Brachychirotherium* and *Isochirotherium*, can be confidently attributed to archosauriforms, based on the presence of a digit IV shorter or as long as digit III. Being metatarsal length directly proportionate to digit length, this assumes that metatarsal IV is shorter than or as long as metatarsal III, a synapomorphy of the archosauriforms (*Nesbitt, 2011*). Other characters useful to identify archosauriforms traces are: (i) the presence of a compact digit group I-IV; (ii) a posterolateral positioned and strongly reduced digit V; (iii) a massive metatarsal-phalangeal region, shorter than or as long as digit I. However, the first character occurs in archosauriforms and non archosauromorphs diapsids (*Haubold, 1971a*; *Haubold, 1971b*; *Smith & Evans, 1996*) whereas the second is present in archosauromorphs (including non-archosauriforms) and lepidosaurs (*Haubold, 1971a*; *Haubold, 1971b*; *Evans & Wang, 2005*; *Gottmann-Quesada & Sander, 2009*). Other features suggesting an archosaur-grade affinity for chirotherian footprints (observed also in the here described traces), are narrow trackways linked to the disposition of limbs under the body, and the presence of small manus relative to the pes, which indicate a possible early tendency toward bipedal posture and gate (see *Haubold, 1971a*; *Haubold, 1971b*; *Haubold, 1984*; *Haubold, 2006*; *Klein et al., 2010*).

To reconstruct the hind- and fore-limb autopodial bones, we assumed an arthral position for the joint articulations within digital pad impressions although aware that this condition is not proven and that the disposition of pads could lie at the phalangeal joints (Fig. 7).

In our opinion, the sub-elliptical to pyriform impression behind group I-IV in *Isochirotherium* could be the result of the coalescence of the impression of the phalangeal-metatarsal portion of digit V and of a thick fleshy pad beneath the astragalus, the calcaneus and some of the tarsal bones. Overall, the trackmaker's pes may have had a semi-plantigrade posture, as evidenced by the gap between digit group I-IV and digit V, corresponding to the part of the foot held up during locomotion. The manus has a marked digitigrade posture and its tetradactyly might result by the fact that manual digit V likely held off the ground during the touch-down and weight bearing phases (*sensu Manning, 2004*).

The reconstructions thus obtained shows the following pes and manus phalangeal formulas: pes 2-3-4-4-1 and manus 1-2-3-3 . They are compared with the anterior and posterior limbs of the main groups of archosauriforms known in the Triassic period (*Von*

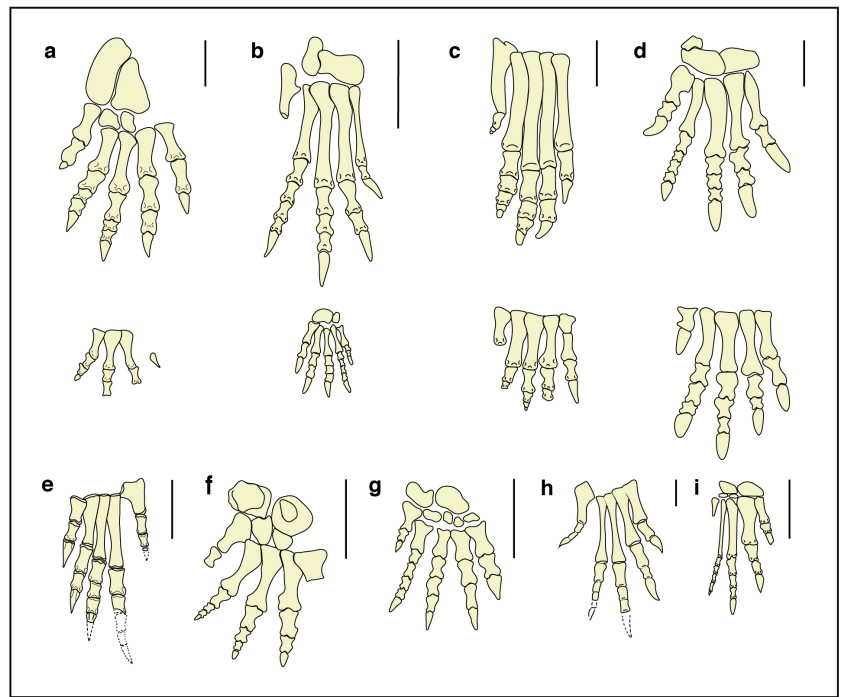

**Figure 9** **Fore- and hind-limb skeletons of Triassic archosauriforms and of the *Isochirotherium gardettensis* trackmaker.** Reconstructed right pes and manus skeletons of (A) the *Isochirotherium gardettensis* trackmaker in anterior/dorsal view; (B) *Postosuchus kirkpatricki*, USA, Norian (redrawn from *Chatterjee, 1985*); (C) *Postosuchus alisonae,* USA, Norian (redrawn from *Peyer et al., 2008*); (D) *Lotosaurus adentus*, China, Ladinian (redrawn from *Zhang, 1975*); (E) *Proterosuchus fergusi*, South Africa, Induan–?early Olenekian (redrawn from *Broom, 1903*); (F) *Erythrosuchus africanus*, South Africa, early Anisian (redrawn from *Broom, 1905*); (G) *Shansisuchus shansisuchus*, China, late Anisian (redrawn from *Young, 1964*); (H) *Euparkeria capensis*, South Africa, Anisian (redrawn from *Broom, 1913*); (I) *Chanaresuchus bonapartei Romer, 1971*, Argentina, Ladinian. Scale bars: (A), (B), (C), (D), (F) and (G) = 10 cm; (E), (H) and (I) = 1 cm.

*Huene, 1902*; *Broom, 1903*; *Broom, 1905*; *Romer, 1971*; *Welles, 1947*; *Young, 1964*; *Zhang, 1975*; *Peyer et al., 2008*; *Ezcurra, Butler & Gower, 2013*; *Sookias & Butler, 2013*; *Trotteyn, Arcucci & Raugust, 2013*).

The first considered non-archosaurian archosauriforms groups are Proterosuchidae (*Ezcurra, Butler & Gower, 2013*), Proterochampsidae (*Trotteyn, Arcucci & Raugust, 2013*) and Euparkeriidae (*Sookias & Butler, 2013*). In all the three representatives *Proterosuchus fergusi* (*Broom, 1903*) (South Africa, Induan–?early Olenekian; Fig. 9E), *Chanaresuchus bonapartei* (*Romer, 1971*) (Argentina, Ladinian; Fig. 9I) and *Euparkeria capensis* (*Broom, 1913*) (South Africa, Anisian; Fig. 9H), the IV metarsal has a length similar or greater than that of the III but the digit II is much shorter than digit III and nearly equal to digit IV, in contrast to what we observe in specimens GT-7-1, GT-7-2 and GT-3. Only disarticulated limb bones are known for the Doswelliidae (*Schoch & Sues, 2014*), another clade of non-archosaurian archosauriforms (Middle-Late Triassic; *Sues, Desojo & Ezcurra, 2013*).

*Diedrich (2015)* recently attributed the *Isochirotherium* tracks to *Arizonasaurus* (*Welles, 1947*), a member of Poposauroidea (Archosauria, Pseudosuchia) found in the Moenkopi Formation (Arizona, USA, Anisian,), from the same levels as *Isochirotherium* tracks. Unfortunately, no bones of the fore- and hind-limbs are known from *Arizonasaurus*, as well as from *Ctenosauriscus koeneni* (*Von Huene, 1902*) (Germany, latest Olenekian), a Lower Triassic poposauroid archosaur, and additionally findings are needed to test Diedrich's hypothesis.

The hind-limb bones are known in *Lotosaurus adentus* (*Zhang, 1975*) (China, Ladinian; Fig. 9D), another member of Poposauroidea with semi-plantigrade posture. If compared with the restored autopodium, it is characterized by larger fore-limbs, digit V positioned further forward, longer metatarsals of digit group I–IV and different digit proportions.

The pedal phalangeal relative length of the "rauisuchid" archosaur *Postosuchus alisonae* (*Peyer et al., 2008*) (USA, Norian; Fig. 9C) is similar but all the five metatarsals are much longer, implying a digitigrade posture, as in the reconstruction proposed by *Peyer et al. (2008)*.

*Postosuchus kirkpatricki* (*Chatterjee, 1985*) (USA, Norian; Fig. 9B), is also characterized by very long metatarsals and thus excluded as a possible trackmaker. The smaller but complete skeleton of *Ticinosuchus ferox* (*Krebs, 1965*) (see *Lautenschlager & Desojo, 2011* for a review of the species) from the uppermost Anisian of Monte San Giorgio (southern Switzerland), shows long metatarsals and a digit IV longer than digit II and is commonly considered as the producer of *Chirotherium* trackways (*Haubold, 1984*; *Haubold, 1986*).

By contrast, the hind limbs of the non-archosaurian archosauriform clade of Erythrosuchidae (*Ezcurra, Butler & Gower, 2013*) are characterized by relative digit length very similar to that outlined for *Isochirotherium gardettensis* and a pedal phalangeal formula that is approximately 2-3-4-5-3 (*Young, 1964*; *Cruickshank, 1978*; *Gower, 1996*).

Metatarsals II and III are sub-equal and slightly longer than IV in *Erythrosuchus africanus* (*Broom, 1905*) (South Africa, lower Anisian; Fig. 9F. See also *Cruickshank, 1978*; *Gower, 1996*).

Metatarsals II and III are the longest in *Shansisuchus shansisuchus* (*Young, 1964*) (Fig. 9G), another member of Erythrosuchidae found in upper Anisian deposits of China; *S. shansisuchus* also possesses a hook-shaped proximal end of metatarsal V and its relative digit proportion closely fits that of our individual, but as for *E. africanus* digit V seems to be too forwardly positioned. However, digit V impression in *I. gardettensis* likely records only the distal metatarsal and phalangeal (ungual) portions. During locomotion the former was held off the ground whereas the latter was likely being retracted due to the presence of a thick fleshy pad beneath calcaneum and astragalus.

The morphology of the acetabulum and proximal end of the femur in erythrosuchids suggests a distinctly sprawling gait (*Gower, 2003*; *Ezcurra, Butler & Gower, 2013*), that clashes with the narrow trackway seen in *I. gardettensis*. Nevertheless, the prominence of metatarsal II and III is evidenced only in non-archosaurian archosauriforms (*Gower, 1996*) and thus an individual belonging to this group, possibly a yet unknown taxon and with a more erect stance and characterized by a marked heteropody, is the most suitable producer (Fig. 10).

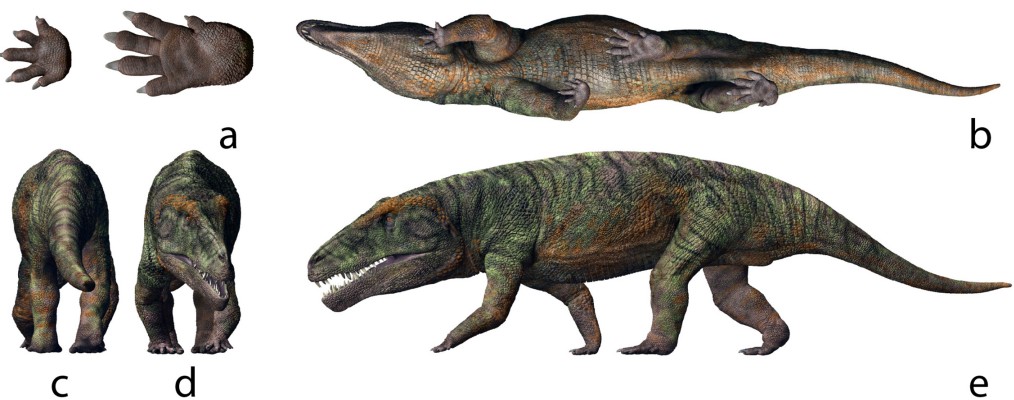

**Figure 10** **Life appearance of the non-archosaurian archosauriform (erythrosuchid?) the most suitable producer of *Isochirotherium gardettensis*.** Simplified reconstruction of fore and hind autopodials in bottom (A) view. Complete life reconstruction in bottom (B), back (C), frontal (D) and lateral view (E) of the trackmaker. The gait and fore- and hind limbs were reconstructed according to the pattern and morphologies of GT-7 trackway (artwork by Fabio Manucci). See the supplementary video to get a more complete view of the reconstruction.

## Biochronology and biogeography

The Gardetta ichnoassemblage represented by *Chirotherium* and *Isochirotherium* is typical for terrestrial deposits of the late Olenekian and early Anisian (*Klein & Haubold, 2007*) and the Gardetta chirotherian tracks correlate with the international *Chirotherium barthii* Assemblage Zone of *Klein & Lucas (2010a)*. This biochron is characterized by the occurrence of *Chirotherium* and *Isochirotherium*, but also by two other ichnogenera not present at Gardetta: *Rotodactylus*, and *Synaptychium*. The *Chirotherium barthii* Assemblage Zone ranges from the late Early to early Middle Triassic (late Olenekian—early Anisian), and independently confirms the Early Triassic (?Olenekian) age, derived by stratigraphic correlation with other sections in the Briançonnais of the Western Alps.

The Gardetta outcrop enlarges also the knowledge on biogeography of archosauriforms in the Lower Triassic of Europe, so far based on archosaur ichnosites discovered in Italy (Val Marenca, *Santi et al., 2015*; Sardinia, *Citton et al., 2020*), Spain (Moncayo and Tagamanent, *Díaz-Martínez & Pérez-García, 2012*), Switzerland (Cascade d'Emaney and Vieux Emosson; *Cavin et al., 2012*), Austria (Drau Range; *Krainer, Lucas & Ronchi, 2012*), Germany (Bundsandstein; *Klein & Haubold, 2007*) and Poland (Wióry, Holy Cross Mountains, *Klein & Niedźwiedzki, 2012*).

Early Triassic erythrosuchid skeletal fossils are known from the late Olenekian of Russia, South Africa, China and India (see *Gower, 2003*; *Ezcurra, Butler & Gower, 2013*; *Ezcurra et al., 2019*; *Ezcurra et al., 2020*; *Gower et al., 2014*; *Ezcurra, 2016*). The Gardetta ichnosite suggests the presence of erythrosuchids and more generally of Archosauriformes at low latitudes (11.8°N) also during the Early Triassic (Fig. 11).
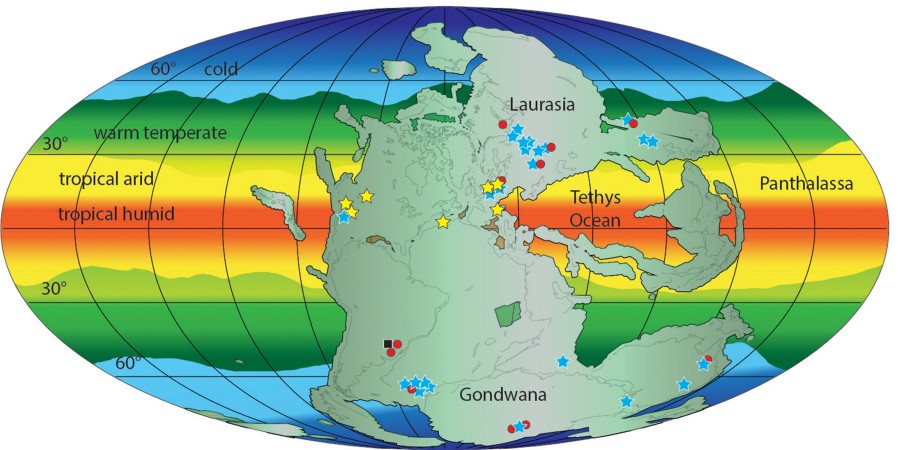

**Figure 11** **Paleogeographic distribution of Early Triassic archosauriform footprints (yellow stars) and body fossil localities across Pangea.** Black square = indeterminate archosauromorphs, red circles = non-archosauriform archosauromorphs, blue stars = archosauriforms. The palaeolatitude estimate for the southern Briançonnais domain is 11.8 N in the Olenekian (250 Ma), confirming that archosauriforms were distributed also at low latitudes, in the tropical humid climatic belt. ImagePaleomap for 250 Ma downloaded from Fossilworks using data from the Paleobiology Database (*Alroy, 2003*). Redrawn and modified from *Bernardi et al. (2015)* and *Benton (2018)*.

# DISCUSSION AND CONCLUSIONS

The Gardetta ichnosite is characterized by archosaur-grade footprints assigned to *Chirotherium* isp. and to the new ichnospecies *Isochirotherium gardettensis*. They represent the first record of terrestrial tetrapods in the Briançonnais domain of the Western Alps and expand the record of archosaur-grade footprints in the Lower Triassic of Central Europe.

The morphological characters of the tracks assigned to *Isochirotherium gardettensis* suggest a non-archosaurian archosauriform (Erythrosuchidae?) as possible trackmaker candidate (even though the presence of crown-archosaurs cannot be excluded), thus providing crucial information about continental tetrapod occurrence in Europe in the Early Triassic. Based on a phylogenetic dataset made by 108 middle Permian–early Late Triassic species, *Ezcurra & Butler (2015)* investigate principal patterns of early archosauromorph biodiversity change across the Permo-Triassic mass extinction. The study, performed using phylogenetic diversity, morphological disparity, number of species and rates of phenotypic evolution across 35 million years of early archosauromorph evolution, indicates consistent phylogenetic diversification of the clade in the Olenekian. In particular, the early diversification of main taxa, which include erythrosuchids, rhynchosaurs and tanystropheids, resulted in significantly high evolutionary rates, with a diversification interpreted by the authors as a radiative response to vacant ecological space, made available by the EPME (*Ezcurra & Butler, 2015*). If the trackmakers' attribution for footprints here described is correct, the material from Gardetta could represent an evidence from Europe of such radiation, with an archosauriform fauna composed at least by ?erythrosuchids (*Isochirotherium gardettensis*) and pseudosuchians (*Chirotherium* isp.). Such clades as
putative trackmaker for the Gardetta tracks are consistent with an Early Triassic (likely late Early Triassic) age, considering that the early history of Archosauriformes is represented essentially by members of Proterosuchidae and Erythrosuchidae (*Charig & Reig, 1970*; *Cruickshank, 1972*; *Charig & Sues, 1976*; *Gower & Sennikov, 2000*; *Gower, 2003*; *Ezcurra, Butler & Gower, 2013*).

Following the huge Permo-Triassic biotic crisis, unfavorable environmental conditions characterized much of the Early Triassic, testifying to one of the slowest recoveries for ecosystems after an extinction in Earth history. A period between five and nine million years for a full recovery has been proposed in several studies (*Hallam, 1991*; *Erwin, 1992*; *Erwin, 2001*; *Payne et al., 2004*; *Payne et al., 2011*; *Algeo et al., 2011*; *Whiteside & Ward, 2011*), inferring a fully restored complex ecosystems only at the beginning of the Middle Triassic (see *Chen & Benton, 2012*). Such long recovery time led to a revolution in both marine and terrestrial ecosystems (*Chen & Benton, 2012*), with a major influence in the evolution of crucial vertebrates clades for the rest of the Mesozoic and Cenozoic eras (*Sepkoski Jr, 1984*; *Benton, 2010*). The recovery period led to the emergence of totally new groups, with a rapid diversification of several lineages of eureptilians both on sea and land (*Nesbitt, Liu & Li, 2010*; *Butler et al., 2011*; *Gower et al., 2014*; *Scheyer et al., 2014*; *Motani et al., 2015a*; *Motani et al., 2015b*; *Peecook, Smith & Sidor, 2018*). Avemetatarsalians (which include dinosaurs and pterosaurs) originated in this period (*Brusatte et al., 2010*; *Nesbitt, Liu & Li, 2010*; *Chen & Benton, 2012*; *Benton, Forth & Langer, 2014*), along with the evolution of crucial modern group ancestors, including crocodiles, lizards, turtles, frogs and mammals. All these aspects highlight the crucial importance of the Early Triassic in the ecosystems restructuring after the Permo-Triassic mass extinction.

*Retallack et al. (2011)* proposed that the long recovery from the mass extinction was strongly influenced by repeated greenhouse crises during the Early Triassic, with consistent negative excursions in carbon isotope ratios indicating at least five greenhouse crises in the 5 Myr following the EPME (Induan-Anisian) (see *Kidder & Worsley, 2004*; *Retallack, 2005*; *Retallack, 2009*; *Retallack, 2013*; *Grasby, Sanei & Beauchamp, 2011*; *Retallack et al., 2011*; *Sun et al., 2012*; *Chen & Benton, 2012*). In this regard, according to *Sun et al. (2012)* the entire Early Triassic was characterized by temperatures consistently in excess of those of the modern equatorial annual sea surface temperatures (SSTs), thus exceeding a tolerable threshold for life in both oceans and on land. Inferring SSTs approaching 40 °C, according to *Sun et al. (2012)* the temperature on land very likely fluctuated to even higher levels, with terrestrial tetrapods generally absent between 30°N and 40°S in the Early Triassic.

In this framework, and although some uncertainties on the chronological attribution persists, the Gardetta ichnosite provides important evidence to the low latitude distribution of archosauriforms during the Early Triassic period, soon after the PTME, corroborating the pattern described by *Bernardi, Petti & Benton (2018)* and *Romano et al. (2020)*. In particular, the new discovery provides further evidence for an early recovery terrestrial ecosystems and the presence at low latitudes of archosauriformes during the Early Triassic. Such evidences support a model in which the EPME did not completely vacate low-latitude lands from tetrapods that, therefore would, have been able to cope with the extreme hot temperatures of Pangaea mainland. In particular, by integrating both skeletal and

ichnological material, recently *Romano et al. (2020)* restricted the "tetrapod gap" of terrestrial life in the Early Triassic to palaeolatitudes between 15°N and about 31°S.

According to *Botha & Smith (2006)*, Archosauromorpha (along with Procolophonomorpha) could be pre-adapted to extremely arid and hot environment conditions, considering that extant reptiles rarely drink water, and are characterized by solute-linked water reabsorption mechanisms, water-resistant integument and low ventilation rates (*Withers, 1992*; *Pough, Heiser & McFarland, 1996*). Such physiological aspects and water conserving mechanisms, probably suggest that response of the archosauriforms to the extreme hot condition of the Early Triassic (*Benton, 2018*) have probably been much more efficient and flexible than previously thought, and did not necessarily imply massive dispersal towards higher latitudes as previously suggested (*Sun et al., 2012*).

Different anatomical features described above indicate the erythrosuchids as the most probable trackmakers for the new described ichnospecies *Isochirotherium gardettensis*. This attribution can also be supported by track parameters such a narrow trackway and high pace angulation, which indicate a more upright posture with respect to a classic plesiomorphic sprawling gait (see *Kubo & Benton, 2007*). In particular, according to *Ezcurra, Butler & Gower (2013)*, erythrosuchids were heavily built and characterized by a probably less sprawling gait, when compared to the condition observed in proterosuchids. The narrow trackway, along with consistently high pace angulation in the Gardetta material, also confirm the statement by *Kubo & Benton (2009)* that, even if proterosuchids and erythrosuchids are traditionally considered as sprawlers, ichnological evidences indicate that archosaurian erect-limb locomotion already evolved in the Early Triassic; the latter conclusion is also supported by ghost ranges from cladograms (*Sereno, 1991*; *Benton, 1999*), and fragmentary materials from Russia (*Gower & Sennikov, 2000*).

To date, erythrosuchids are totally unknown from North America and Europe, being described only from Russia, South Africa, China and India (see *Gower, 2003*; *Ezcurra, Butler & Gower, 2013*; *Ezcurra et al., 2019*; *Ezcurra et al., 2020*; *Gower et al., 2014*; *Ezcurra, 2016*). Thus, the material from the Lower Triassic deposits of Gardetta could represent the first occurrence of the clade in Europe, although, as already pointed out, the attribution is for the moment only tentative and new studies are underway to better constrain the identity of the zoological trackmaker.

The planned future excavations in the Gardetta ichnosite will hopefully provide additional data to improve our knowledge of the evolutionary history of Archosauriformes in the aftermath of the EPME.

**Institutional abbreviations**

**MGPT-PU**    Museo di Geologia e Paleontologia dell' Università di Torino, Italy

## ACKNOWLEDGEMENTS

We thank the reviewers Ignacio Díaz-Martínez, Heitor Francischini, Martin Lockley and the Editor Graciela Piñeiro for their corrections and suggestions that greatly improved the manuscript. A special thanks to Giovanni Raggi for his valuable and constant support during

the field works and the project organisation. We acknowledge insightful discussions with A. d'Atri (University of Torino). The authors wish also to thank Dr. Debora Rocchietti and Dr. Alberto Crosetto (Soprintendenza Archeologia Belle Arti e Paesaggio per le province di Alessandria, Asti e Cuneo) and Dr. Attilio Dalmasso (Museo dei fossili in San Rocco di Bernezzo) for their assistance. Fabio Manucci is thanked for video production and artwork. Finally, a special thank is also due to Hanna Luginbühl for her help in mapping 2009 and to Cecilia Gomiero, Jacopo Valori and Nicolò Amoruso for their precious help during 2018 fieldwork. This is the publication number 349 of the Museo di Geologia e Paleontologia collections at the Università degli Studi di Torino.

### Funding

This work was supported by the Cultural Association Escarton (funds awarded to Dr. Giovanni Raggi), and the Euregio Science Fund (call 2014, IPN16) of the Europaregion Euregio within the project 'The end-Permian mass extinction in the Southern and Eastern Alps: extinction rates versus taphonomic biases in different depositional environments'. There was no additional external funding received for this study. The funders had no role in study design, data collection and analysis, decision to publish, or preparation of the manuscript.

### Grant Disclosures

The following grant information was disclosed by the authors:
Cultural Association Escarton.
Euregio Science Fund.

### Competing Interests

The authors declare there are no competing interests.

### Author Contributions

- Fabio Massimo Petti conceived and designed the experiments, performed the experiments, analyzed the data, prepared figures and/or tables, authored or reviewed drafts of the paper, and approved the final draft.
- Heinz Furrer, Enrico Collo, Edoardo Martinetto, Massimo Bernardi, Massimo Delfino, Marco Romano and Michele Piazza conceived and designed the experiments, performed the experiments, analyzed the data, authored or reviewed drafts of the paper, and approved the final draft.

### Field Study Permissions

The following information was supplied relating to field study approvals (i.e., approving body and any reference numbers):

The field season 2008, 2009, 2017 and 2018 were exclusively based on surface surveys that according to the Italian law do not require any permit, in total agreement with the regional office Soprintendenza Archeologia, Belle Arti e Paesaggio per le province di Alessandria, Asti e Cuneo.

## Data Availability
The raw data are represented only by footprint measurements and trackway parameters fully reported in the Description section of the article.

The described material is represented by a footprint still in situ. Three solitary small footprints not from the holotype were collected from loose slabs and were temporarily stored at Museo di Geologia e Paleontologia dell'Università di Torino.

The printed 3D model (MGPT-PU135785): of the holotype is stored in the collection of the Museo di Geologia e Paleontologia dell'Università di Torino (Museum of Geology and Paleontology of the Turin University) and three isolated small footprints (GD-E1, GD-E2, GD-12), preserved as convex epirelief of the directly overlying sandstone bed, collected from loose slabs, are currently stored at the Museo di Geologia e Paleontologia dell'Università di Torino (Turin, Italy).

The 3D model of the holotype is digitally stored in MorphoSource: https://www.morphosource.org/Detail/MediaDetail/Show/media_id/85469, DOI: 10.17602/M2/M166068.

## New Species Registration
The following information was supplied regarding the registration of a newly described species:

Publication LSID: urn:lsid:zoobank.org:pub:654063D7-8AE9-4F4A-A6E9-E518F3D6B79E

*Isochirotherium gardettensis* ichnosp. nov. LSID: urn:lsid:zoobank.org:act:10B02112-7576-4135-B600-816480DAF658

## Supplemental Information
Supplemental information for this article can be found online at http://dx.doi.org/10.7717/peerj.10522#supplemental-information.

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

# PeerJ

**Avanzini M, Lockley M. 2002.** Middle Triassic archosaur population structure: interpretation based on *Isochirotherium delicatum* fossil footprints (Southern Alps, Italy). *Palaeogeography, Palaeoclimatology, Palaeoecology* **185(3-4)**:391–402 DOI 10.1016/S0031-0182(02)00441-8.

**Avanzini M, Mietto P. 2008.** Lower and Middle Triassic footprint-based biochronology in the Italian Southern Alps. *Oryctos* **8**:3–13.

**Baird D. 1954.** *Chirotherium lulli*, a pseudosuchian reptile from New Jersey. *Museum of Comparative Zoology Bulletin* **111**:165–192.

**Baud A. 1976.** Les terriers de Crustacés décapodes et l'origine de certains facies du Trias carbonaté. *Eclogae Geologicae Helvetiae* **69(2)**:415–424.

**Baud A, Plasencia P, Hirsch F, Richoz S. 2016.** Revised middle Triassic stratigraphy of the Swiss Prealps based on conodonts and correlation to the Briançonnais (Western Alps). *Swiss Journal of Geosciences* **109**:365–377 DOI 10.1007/s00015-016-0226-3.

**Belvedere M, Farlow JO. 2016.** A numerical scale for quantifying the quality of preservation of vertebrate tracks. In: Falkingham PL, Marty D, Richter A, eds. *Dinosaur tracks: the next steps.* Bloomington: Indiana University Press, 92–98.

**Benton MJ. 1999.** Scleromochlus taylori and the origin of dinosaurs and pterosaurs. *Philosophical Transactions of the Royal Society of London. Series B: Biological Sciences* **354(1388)**:1423–1446 DOI 10.1098/rstb.1999.0489.

**Benton MJ. 2003.** *When life nearly died: the greatest mass extinction of all time.* London: Thames & Hudson.

**Benton MJ. 2010.** The origins of modern biodiversity on land. *Philosophical Transactions of the Royal Society B: Biological Sciences* **365(1558)**:3667–3679 DOI 10.1098/rstb.2010.0269.

**Benton MJ. 2018.** Hyperthermal-driven mass extinctions: killing models during the Permian–Triassic mass extinction. *Philosophical Transactions of the Royal Society A* **376**:20170076 DOI 10.1098/rsta.2017.0076.

**Benton MJ, Forth J, Langer MC. 2014.** Models for the rise of the dinosaurs. *Current Biology* **24**:R87–R95 DOI 10.1016/j.cub.2013.11.063.

**Benton M, Newell AJ. 2014.** Impacts of global warming on Permo-Triassic terrestrial ecosystems. *Gondwana Research* **25**:1308–1337 DOI 10.1016/j.gr.2012.12.010.

**Benton MJ, Twitchett RJ. 2003.** How to kill (almost) all life: the end–Permian extinction event. *Trends in Ecology & Evolution* **18**:358–365 DOI 10.1016/S0169-5347(03)00093-4.

**Bernardi M, Klein H, Petti FM, Ezcurra MD. 2015.** The origin and early radiation of archosauriforms: integrating the skeletal and footprint record. *PLOS ONE* **10(6)**:e0128449 DOI 10.1371/journal.pone.0128449.

**Bernardi M, Petti FM, Benton MJ. 2018.** Tetrapod distribution and temperature rise during the Permian—Triassic mass extinction. *Proceedings of the Royal Society of London B* **285**:20172331.

**Berner RA. 2002.** Examination of hypotheses for the Permo–Triassic boundary extinction by carbon cycle modeling. *Proceedings of the National Academy of Sciences of the United States of America* **99**:4172–4177 DOI 10.1073/pnas.032095199.

**Bertok C, Martire L, Perotti E, D'Atri A, Piana F. 2012.** Kilometre-scale palaeoescarpments as evidence for Cretaceous synsedimentary tectonics in the External Briançonnais Domain (Ligurian Alps, Italy). *Sedimentary Geology* **251**:58–75.

**Botha J, Smith RMH. 2006.** Rapid vertebrate recuperation in the Karoo Basin of South Africa following the end–Permian extinction. *Journal of African Earth Sciences* **45**:502–514 DOI 10.1016/j.jafrearsci.2006.04.006.

**Broom R. 1903.** On a new reptile (*Proterosuchus fergusi*) from the Karroo beds of Tarkastad, South Africa. *Annals of the South African Museum* **4**:159–164.

**Broom R. 1905.** Notice of some new reptiles from the Karroo Beds of South Africa. *Records of the Albany Museum* **1**:331–337.

**Broom R. 1913.** Note on *Mesosuchus browni*, Watson, and on a new South African Triassic pseudosuchian (*Euparkeria capensis*). *Records of the Albany Museum* **2**:394–396.

**Brusatte SL, Benton MJ, Desojo JB, Langer MC. 2010.** The higher-level phylogeny of Archosauria (Tetrapoda: Diapsida). *Journal of Systematic Palaeontology* **8(1)**:3–47 DOI 10.1080/14772010903537732.

**Butler RJ, Brusatte SL, Reich M, Nesbitt SJ, Schoch RR, Hornung JJ. 2011.** The sail–backed reptile *Ctenosauriscus* from the latest Early Triassic of Germany and the timing and biogeography of the early archosaur radiation. *PLOS ONE* **6(10)**:e25693 DOI 10.1371/journal.pone.0025693.

**Caby R, Galli J. 1964.** Existence de cinérites et tufs volcaniques dans le Trias moyen de la zone briançonnaise. *Comptes Rendus de l'Académie des Sciences de Paris* **259**:417–420.

**Carrano MT, Wilson JA. 2001.** Taxon distributions and the tetrapod track record. *Paleobiology* **27(3)**:564–582 DOI 10.1666/0094-8373(2001)027<0564:TDATTT>2.0.CO;2.

**Carraro F, Dal Piaz GV, Franceschetti B, Malaroda R, Sturani C, Zanella E. 1970.** Carta Geologica del massiccio dell'Argentera alla scala 1: 50.000 e Note Illustrative. *Memorie della Società Geologica Italiana* **9**:557–663.

**Cassinis G, Perotti C, Santi G. 2018.** Post-Variscan Verrucano-like deposits in Italy, and the onset of the alpine tectono-sedimentary cycle. *Earth-Science Reviews* **185**:476–497 DOI 10.1016/j.earscirev.2018.06.021.

**Cavin L, Avanzini M, Bernardi M, Piuz A, Proz PA, Meister C, Boissonnas J, Meyer CA. 2012.** New vertebrate trackways from the autochthonous cover of the Aiguilles Rouges Massif and reevaluation of the dinosaur record in the Valais, SW Switzerland. *Swiss Journal of Palaeontology* **131**:317–324 DOI 10.1007/s13358-012-0040-0.

**Charig AJ, Reig OA. 1970.** The classification of the Proterosuchia. *Biological Journal of the Linnean Society* **2(2)**:125–171 DOI 10.1111/j.1095-8312.1970.tb01708.x.

**Charig AJ, Sues H-D. 1976.** Proterosuchia. In: Kuhn O, ed. *Handbuch der Paläoherpetologie*. 13. Stuttgart: Gustav Fischer, 11–39.

Chatterjee S. 1985. Postosuchus, a new thecodontian reptile from the Triassic of Texas and the origin of tyrannosaurs. *Philosophical Transactions of the Royal Society of London B* **309**:395–460 DOI 10.1098/rstb.1985.0092.

Chen ZQ, Benton MJ. 2012. The timing and pattern of biotic recovery following the end-Permian mass extinction. *Nature Geoscience* **5(6)**:375–383 DOI 10.1038/ngeo1475.

Citton P, Ronchi A, Nicosia U, Sacchi E, Maganuco S, Cipriani A, Innamorati G, Zuccari C, Manucci F, Romano M. 2020. Tetrapod tracks from the Middle Triassic of NW Sardinia (Nurra region, Italy). *Italian Journal of Geosciences* **139(2)**:309–320 DOI 10.3301/IJG.2020.07.

Costamagna LG. 2013. Middle Triassic carbonate lithostratigraphy of the Southern Briançonnais (Cottian Alps, Italy) and comparison with the surrounding areas. *GeoActa* **12**:1–24.

Costamagna LG, Barca S, Nervo R. 2002. Analisi di facies della successione carbonatica mediotriassica del Dominio Brianzonese fra le valli Stura e Maira (Alpi Occidentali, Cuneo, Italia): La sezione del Vallone del Preit. In: Fioraso G, Malusà M, Mosca P, Tallone S, eds. *81a Riunione estiva SGI, Riassunti delle Comunicazioni orali e dei poster*. 110–111.

Courel L, Demathieu G. 1976. Une ichnofaune reptilienne remarquable dans les grès Triasique de Largentière (Ardèche, France). *Palaeontographica A* **151**:194–216.

Courel L, Demathieu G, Gall JC. 1979. Figures sédimentaires et traces d'origine bi-ologique du Trias moyen de la bordure orientale du Massif Central. Signification sédimentologique et paleoécologique. *Geobios* **12**:379–397 DOI 10.1016/S0016-6995(79)80118-7.

Cruickshank ARI. 1972. The proterosuchian thecodonts. In: Joysey KA, Kemp TS, eds. *Studies in Vertebrate Evolution*. Edinburgh: Oliver and Boyd, 89–119.

Cruickshank ARI. 1978. The pes of *Eythrosuchus africanus* Broom. *Zoological Journal of the Linnean Society* **62**:161–177 DOI 10.1111/j.1096-3642.1978.tb01035.x.

D'Atri A, Piana F, Barale L, Bertok C, Martire L. 2016. Geological setting of the southern termination of Western Alps. *International Journal of Earth Sciences* **105(6)**:1831–1858 DOI 10.1007/s00531-015-1277-9.

Decarlis A, Dallagiovanna G, Lualdi A, Maino M, Seno S. 2013. Stratigraphic evolution in the Ligurian Alps between Variscan heritages and the Alpine Tethys opening: a review. *Earth-Science Reviews* **125**:43–68 DOI 10.1016/j.earscirev.2013.07.001.

Decarlis A, Lualdi A. 2009. A sequence stratigraphic approach to a Middle Triassic shelf-slope complex of the Ligurian Alps (Ligurian Briançonnais, Monte Carmo-Rialto unit, Italy). *Facies* **55**:267–290 DOI 10.1007/s10347-008-0170-4.

Decarlis A, Manatschal G, Haupert I, Masini E. 2015. The tectono-stratigraphic evolution of distal, hyper-extended magma-poor conjugate rifted margins: examples from the Alpine Tethys and Newfoundland–Iberia. *Marine and Petroleum Geology* **68**:54–72 DOI 10.1016/j.marpetgeo.2015.08.005.

Demathieu G. 1970. Les empreintes de pas de vertébrés du Trias de la bordure Nord-Est du Massif Central. In: *Cahiers de Paléontologie*. Paris: CRNS, 211 p..

Demathieu G. 1984. Une ichnofaune du Trias moyen du basin de Lodève (Hérault, France). *Annales de Palontologie (Vertebrates-Invertebrates)* **70**:247–273.

Demathieu G, Weidmann M. 1982. Les empreintes de pas de reptiles dans le Trias du Vieux Emosson (Finhaut, Valais, Suisse). *Eclogae Geologicae Helvetiae* **75**:721–757.

Dìaz-Martinez I, Castanera D, Gasca JM, Canudo JI. 2015. A reappraisal of the Middle Triassic chirotheriid Chirotherium ibericus Navas, 1906 (Iberian Range, NE Spain), with comments on the Triassic tetrapod track biochronology of the Iberian Peninsula. *PeerJ* **3**:1044 DOI 10.7717/peerj.1044.

Díaz-Martínez I, Pérez-García A. 2012. Historical and comparative study of the first Spanish vertebrate paleoichnological record and bibliographic review of the Spanish chirotheriid footprints. *Ichnos* **19(3)**:141–149 DOI 10.1080/10420940.2012.685565.

Diedrich C. 2015. Isochirotherium trackways, their possible trackmakers (?*Arizonasaurus*): intercontinental giant archosaur migrations in the Middle Triassic tsunami-influenced carbonate intertidal mud flats of the European Germanic Basin. *Carbonates and Evaporites* **30**:229–252 DOI 10.1007/s13146-014-0228-z.

Egerton PG. 1839. On two casts in sandstone of impression of the Hind Foot of a gigantic *Cheirotherium*, from the New red sandstone of Cheshire. *The London and Edinburgh Philosophical Magazine and Journal of Science, 3rd series* **14(75)**:151–158.

Erwin DH. 1992. A preliminary classification of evolutionary radiations. *Historical Biology* **6**:133–147 DOI 10.1080/10292389209380423.

Erwin DH. 1993. *The Great paleozoic crisis, life and death in the Permian*. New York: Colombia University Press, 327 pp.

Erwin DH. 2001. Lessons from the past: biotic recoveries from mass extinctions. *Proceedings of the National Academy of Sciences of the United States of America* **98(10)**:5399–5403 DOI 10.1073/pnas.091092698.

Evans SE, Wang Y. 2005. Dalinghosaurus, a lizard from the early Cretaceous Jehol Biota of north- east China. *Acta Paleontologica Polonica* **50**:725–742.

Ezcurra MD. 2016. The phylogenetic relationships of basal archosauromorphs, with an emphasis on the systematics of proterosuchian archosauriforms. *PeerJ* **4**:e1778 DOI 10.7717/peerj.1778.

Ezcurra MD, Butler RJ. 2015. Taxonomy of the proterosuchid archosauriforms (Diapsida: Archosauromorpha) from the earliest Triassic of South Africa, and implications for the early archosauriform radiation. *Palaeontology* **58(1)**:141–170 DOI 10.1111/pala.12130.

Ezcurra MD, Butler RJ, Gower DJ. 2013. 'Proterosuchia': the origin and early history of Archosauriformes. *Geological Society, London, Special Publications* **379(1)**:9–33 DOI 10.1144/SP379.11.

Ezcurra MD, Gower DJ, Sennikov AG, Butler RJ. 2019. The osteology of the holotype of the early erythrosuchid Garjainia prima Ochev, 1958 (Diapsida: Archosauromorpha)

from the upper Lower Triassic of European Russia. *Zoological Journal of the Linnean Society* **185**:717–783 DOI 10.1093/zoolinnean/zly061.

**Ezcurra MD, Jones AS, Gentil AR, Butler RJ. 2020.** Early Archosauromorphs: the Crocodile and Dinosaur precursors. Encyclopedia of geology. In: *Reference Module in Earth Systems and Environmental Sciences*. 2nd edition Amsterdam, The Netherlands: Elsevier.

**Feldmann M, Furrer H, Glarus K. 2009.** Die Saurierspuren am Tödi und ihre geologische Umgebung. *Mitteilungen der Naturforschenden Gesellschaft des Kantons Glarus* **18**:28–37.

**Fortuny J, Bolet A, Selles AG, Cartanya J, Galobart A. 2011.** New insights on the Permian and Triassic vertebrates from the Iberian peninsula with emphasis on the Pyrenean and Catalonian basins. *Journal of Iberian Geology* **37(1)**:65–86.

**Gand G. 1978.** Interprètations palèontologique et palèoècologique d'un sixième assemblage à traces de reptiles des carrières triasiques de St.-Sernin-du-Bois (Autunois, France). Conclusions gènèrales à ètude du gisement. *Bulletin de la Société d'Historie Naturelle d'Autun* **87**:9–29.

**Gand G. 1979.** Description de deux nouvelles traces d'*Isochirotherium* observées dans les grès du Trias moyen de Bourgogne. *Bulletin de la Socit d'Histoire Naturelle Creusot* **37**:13–25.

**Gidon M. 1958a.** Nouvelles observations sur la zone briançonnaise au delà de la frontiére franco-italienne (Bassin de la Haute Maira, Province de Cuneo). *Trav. Laboratory geology Universite Grenoble* **34**:153–167.

**Gidon M. 1958b.** PhD Thesis, Faculté des Sciences de l'Université de Grenoble, 272 pp.

**Gidon M. 1972.** Les chainons briançonnais et subbriançonnais de la rive gauche de la Stura entre la Val de l'Arma (province de Cuneo-Italie). *Géologie Alpine* **48(1)**:87–120.

**Gidon M. 1978.** Carte géologique détaillée de la France à l'échelle 1/50.000, feuille Larche, 1° édition. Bureau de Recherche Géologique et Minière, Orléans, with explanatory notes. 1–28.

**Golden Software. 2002.** Surfer version 8.0: surface mapping system.

**Gottmann-Quesada A, Sander PM. 2009.** A redescription of the early archosauromorph *Protorosaurus speneri* Meyer, 1832 and its phylogenetic relationships. *Palaeontographica Abteilung A* **287**:123–220 DOI 10.1127/pala/287/2009/123.

**Gower DJ. 1996.** The tarsus of erythrosuchid archosaurs, and implications for early diapsid phylogeny. *Zoological Journal of the Linnean Society* **116(4)**:347–375 DOI 10.1111/j.1096-3642.1996.tb00128.x.

**Gower DJ. 2003.** Osteology of the early archosaurian reptile *Erythrosuchus africanus* Broom. *Annals of the South African Museum* **110**:1–88.

**Gower DJ, Hancox PJ, Botha-Brink J, Sennikov AG, Butler RJ. 2014.** A new species of *Garjainia* Ochev, 1958 (Diapsida: Archosauriformes: Erythrosuchidae) from the Early Triassic of South Africa. *PLOS ONE* **9(11)**:e111154 DOI 10.1371/journal.pone.0111154.

**Gower DJ, Sennikov AG. 2000.** Early Archosaurs from Russia. In: Benton MJ, Shishkin MA, Unwin DM, Kurochkin EN, eds. *The age of Dinosaurs in Russia and Mongolia.* Cambridge: Cambridge University Press, 140–159.

**Grasby SE, Sanei H, Beauchamp B. 2011.** Catastrophic dispersion of coal fly ash into oceans during the latest Permian extinction. *Nature Geoscience* **4(2)**:104–107 DOI 10.1038/ngeo1069.

**Hallam A. 1991.** Why was there a delayed radiation after the end-Palaeozoic extinctions? *Historical Biology* **5(2-4)**:257–262 DOI 10.1080/10292389109380405.

**Haubold H. 1967.** Eine Pseudosuchier- Fährtenfauna aus dem buntsandstein südthüringens. *Hallesches Jahrbuch für Mitteldeutsche Erdgeschichte* **8**:12–48.

**Haubold H. 1971a.** Die Tetrapodenfährten des Buntsandsteins. *Paläontologische Abhandlungen A* **4(3)**:395–548.

**Haubold H. 1971b.** Ichnia Amphibiorum et Reptiliorum fossilium. *Encyclopedia of Paleoherpetology* **18**:1–124.

**Haubold H. 1984.** *Saurierfährten.* Wittenberg: Ziemsen, 231 p.

**Haubold H. 1986.** Archosaur footprints at the terrestrial Triassic–Jurassic transition. In: Padian K, ed. *The beginning of the age of Dinosaurs: faunal change across the Triassic–Jurassic boundary.* Cambridge: Cambridge University Press, 190–201.

**Haubold H. 2006.** Die Saurierfährten *Chirotherium barthii* (*Kaup, 1835*)—das Typusmaterial aus dem Buntsandstein bei Hildburghausen/Thüringen und das Chirotherium-Monument. *Veröffentlichungen Naturhistorisches Museum Schleusingen* **21**:3–31.

**Haubold H, Klein H. 2002.** Chirotherien und Grallatoriden aus der Unteren bis Oberen Trias Mitteleuropas und die Entstehung der Dinosauria. *Hallesches Jahrbuch für Geowissenschaften B* **24**:1–22.

**Joachimski MM, Lai X, Shen S, Jiang H, Luo G, Chen B, Chen J, Sun Y. 2012.** Climate warming in the latest Permian and the Permian–Triassic mass extinction. *Geology* **40**:195–198 DOI 10.1130/G32707.1.

**Kaup JJ. 1835.** Über Tierfährten bei Hildburghausen. *Neues Jahrbuch für Mineralogie, Geologie und Paläontologie* **1835**:327–328.

**Kidder DL, Worsley TR. 2004.** Causes and consequences of extreme Permo–Triassic warming to globally equable climate and relation to the Permo–Triassic extinction and recovery. *Palaeogeography, Palaeoclimatology, Palaeoecology* **203(3–4)**:207–237 DOI 10.1016/S0031-0182(03)00667-9.

**King MJ, Sarjeant WAS, Thompson DB, Tresise G. 2005.** A revised systematic ichnotaxonomy and review of the vertebrate footprint ichnofamily Chirotheriidae from the British Triassic. *Ichnos* **12**:241–299 DOI 10.1080/10420940591009312.

**Klein H, Haubold H. 2007.** Archosaur footprints-potential for biochronology of Triassic continental sequences. *New Mexico Museum of Natural History and Science Bulletin* **41**:120–130.

**Klein H, Lucas SG. 2010a.** Tetrapod footprints and their use in biostratigraphy and biochronology of the Triassic. In: Lucas SG, ed. *The Triassic timescale.* 334. London: Geological Society of London Special Publications, 419–446.

**Klein H, Lucas SG. 2010b.** Review of the tetrapod ichnofauna of the Moenkopi Formation/Group (Early-Middle Triassic) of the American Southwest. *New Mexico Museum of Natural History and Science Bulletin* **50**:1–67.

**Klein H, Niedźwiedzki G. 2012.** Revision of the Lower Triassic tetrapod ichnofauna from Wióry, Holy Cross Mountains, Poland. *New Mexico Museum of Natural History and Science, Bulletin* **56**:1–62.

**Klein H, Voigt S, Hminna A, Saber H, Schneider J, Hmich D. 2010.** Early Triassic archosaur-dominated footprint assemblage from the Argana Basin (western High Atlas, Morocco). *Ichnos* **17(3)**:215–227 DOI 10.1080/10420940.2010.510030.

**Klein H, Voigt S, Saber H, Schneider JW, Hminna A, Fischer J, Lagnaoui A, Brosig A. 2011.** First occurrence of a Middle Triassic tetrapod ichnofauna from the Argana Basin (Western High Atlas, Morocco). *Palaeogeography, Palaeoclimatology, Palaeoecology* **307**:218–231 DOI 10.1016/j.palaeo.2011.05.021.

**Klein H, Wizevich MC, Thüring B, Marty D, Thüring S, Falkingham P, Meyer CA. 2016.** Triassic chirotheriid footprints from the Swiss Alps: ichnotaxonomy and depositional environment (Cantons Wallis & Glarus). *Swiss Journal of Palaeontology* **135(2)**:295–314 DOI 10.1007/s13358-016-0119-0.

**Knoll AH, Bambach RK, Payne JL, Pruss S, Fischer WW. 2007.** Paleophysiology and end–Permian mass extinction. *Earth and Planetary Science Letters* **256(3–4)**:295–313 DOI 10.1016/j.epsl.2007.02.018.

**Krainer K, Lucas SG, Ronchi A. 2012.** Tetrapod footprints from the Alpine Buntsandstein (Lower Triassic) of the Drau Range (Eastern Alps, Austria). *Jahrbuch der Geologischen Bundesanstalt* **152**:205–212.

**Krebs B. 1965.** Die Triasfauna der Tessiner Kalkalpen, XIX. Ticinosuchus ferox nov. gen. nov. sp. *Schweizerische Paläontologische Abhandlungen* **81**:1–140.

**Kubo T, Benton MJ. 2007.** Evolution of hindlimb posture in archosaurs: limb stresses in extinct vertebrates. *Palaeontology* **50(6)**:1519–1529 DOI 10.1111/j.1475-4983.2007.00723.x.

**Kubo T, Benton MJ. 2009.** Tetrapod postural shift estimated from Permian and Triassic trackways. *Palaeontology* **52(5)**:1029–1037 DOI 10.1111/j.1475-4983.2009.00897.x.

**Lautenschlager S, Desojo JB. 2011.** Reassessment of the Middle Triassic rauisuchian archosaurs *Ticinosuchus ferox* and *Stagonosuchus nyassicus*. *Paläontologische Zeitschrift* **85(4)**:357–381 DOI 10.1007/s12542-011-0105-1.

**Leonardi G. 1987.** *Glossary and manual of tetrapod footprint palaeoichnology*. Brasilia: Ministerio das Minas e Energia Departamento Nacional da Producao Mineral, 117pp.

**Lorenzoni S, Zanettin E. 1958.** Contributo alla conoscenza del giacimento uranifero di Preit (Alpi Cozie). *Studi e Ricerche Divisione Geomineraria CNRN* **1(2)**:349–433.

**Lualdi A, Bianchi U. 1990.** La Formazione di Costa Losera: una nuova unità stratigrafica dell'Anisico delle Alpi Liguri. *Atti Ticinensi di Scienze della Terra* **33**:33–62.

**Lualdi A, Seno S. 1984.** Osservazioni stratigrafiche e tettoniche nella zona dei Rio di Nava (Brianzonese Ligure Esterno, Unità di Ormea). *Memorie della Società Geologica Italiana* **28**:493–503.

**Malaroda R. 1970.** Carta geologica del Massiccio dell'Argentera alla scala 1: 50,000. *Memorie della Società Geologica Italiana* **9**:557–663.

**Mallison H, Wings O. 2014.** Photogrammetry in paleontology—a practical guide. *Journal of Paleontological Techniques* **12**:1–31.

**Manning PL. 2004.** A new approach to the analysis and interpretation of tracks: examples from the dinosauria. *Geological Society, London, Special Publications* **228(1)**:93–123 DOI 10.1144/GSL.SP.2004.228.01.06.

**Megard-Galli J, Baud A. 1977.** Le Trias moyen et supérieur des Alpes nord-occidentales et occidentales: données nouvelles et corrélations stratigraphiques. *Bulletin B.R.G.M* **4(3)**:233–250.

**Melchor RN, De Valais S. 2006.** A review of Triassic tetrapod track assemblages from Argentina. *Palaeontology* **49(2)**:355–379 DOI 10.1111/j.1475-4983.2006.00538.x.

**Motani R, Jiang DY, Chen GB, Tintori A, Rieppel O, Ji C, Huang JD. 2015a.** A basal ichthyosauriform with a short snout from the Lower Triassic of China. *Nature* **517(7535)**:485–488 DOI 10.1038/nature13866.

**Motani R, Jiang DY, Tintori A, Rieppel O, Chen GB, You H. 2015b.** Status of *Chaohusaurus chaoxianensis* (Chen, 1985). *Journal of Vertebrate Paleontology* **35(1)**:e892011 DOI 10.1080/02724634.2014.892011.

**Nesbitt SJ. 2011.** The early evolution of archosaurs: relationships and the origin of major clades. *Bulletin of the American Museum of Natural History* **41**:1–292.

**Nesbitt SJ, Liu J, Li C. 2010.** A sail-backed suchian from the Heshanggou Formation (Early Triassic: Olenekian) of China. *Earth and Environmental Science Transactions of the Royal Society of Edinburgh* **101**:271–284 DOI 10.1017/S1755691011020044.

**Olsen PE. 1995.** A new approach for recognizing track makers. *Geological Society of America, Abstracts with Programs* **27**:72.

**Olsen PE, Smith JB, McDonald NG. 1998.** Typematerial of the type species of the classic theropod footprint genera Eubrontes, Anchisauripus and Grallator (Early Jurassic, Hartford and Deerfield basins, Connecticut and Massachusetts, U.S.A.). *The Journal of Vertebrate Paleontology* **18**:586–601 DOI 10.1080/02724634.1998.10011086.

**Payne JL, Lehrmann DJ, Wei J, Orchard MJ, Schrag DP, Knoll AH. 2004.** Large perturbations of the carbon cycle during recovery from the end-Permian extinction. *Science* **305**:506–509 DOI 10.1126/science.1097023.

**Payne JL, Summers M, Rego BL, Altiner D, Wei J, Yu M, Lehrmann DJ. 2011.** Early and Middle Triassic trends in diversity, evenness, and size of foraminifers on a carbonate platform in south China: implications for tempo and mode of biotic recovery from the end-Permian mass extinction. *Paleobiology* **37**:409–425 DOI 10.1666/08082.1.

**Peabody FE. 1948.** Reptile and amphibian trackways from the Lower Triassic Moenkopi formation of Arizona and Utah. *Bulletin of the Department of Geological sciences* **27**:295–468.

**Peecook BR, Smith RM, Sidor CA. 2018.** A novel archosauromorph from Antarctica and an updated review of a high-latitude vertebrate assemblage in the wake of the end-Permian mass extinction. *Journal of Vertebrate Paleontology* **38(6)**:e1536664 DOI 10.1080/02724634.2018.1536664.

**Petti FM, Avanzini M, Belvedere M, De Gasperi M, Ferretti P, Girardi S, Remondino F, Tomasoni R. 2008.** Digital 3D modelling of dinosaur footprints by photogrammetry and laser scanning techniques: integrated approach at the Coste dell'Anglone tracksite (Lower Jurassic, Southern Alps, Northern Italy). *Studi Trentini di Scienze Naturali, Acta Geologica* **83**:303–315.

**Petti FM, Bernardi M, Kustatscher E, Renesto S, Avanzini M. 2013.** Diversity of continental tetrapods and plants in the Triassic of the Southern Alps: Ichnological, paleozoological and paleobotanical evidence. In: Tanner LH, Spielmann JA, Lucas SG, eds. *The Triassic system*. 61. New Mexico Museum of Natural History and Science, Bulletin, 458–484.

**Peyer K, Carter JG, Sues H-D, Novak SE, Olsen PE. 2008.** A new suchian archosaur from the Upper Triassic of North Carolina. *Journal of Vertebrate Paleontology* **28**:363–381 DOI 10.1671/0272-4634(2008)28[363:ANSAFT]2.0.CO;2.

**Pough FH, Heiser JB, McFarland WN. 1996.** *Vertebrate life*. New Jersey: Prentice Hall International.

**Racki G. 2003.** End–Permian mass extinction: oceanographic consequences of double catastrophic volcanism. *Lethaia* **35**:171–173.

**Racki G, Wignall PB. 2005.** Late Permian double–phased mass extinction and volcanism: an oceanographic perspective. In: Over DJ, Morrow JR, Wignall PB, eds. *Understanding late Devonian and Permian–Triassic biotic and climatic events: towards an integrated approach*. Elsevier B.V. 263–297.

**Remondino F, Rizzi A, Girardi S, Petti FM, Avanzini M. 2010.** 3D Ichnology— recovering digital 3D models of dinosaur footprints. *The Photogrammetric Record* **25(131)**:266–282 DOI 10.1111/j.1477-9730.2010.00587.x.

**Retallack GJ. 2005.** Permian greenhouse crises. The nonmarine Permian. *New Mexico Museum of Natural History and Science Bulletin* **30**:256–269.

**Retallack GJ. 2009.** Greenhouse crises of the past 300 million years. *Geological Society of America Bulletin* **121(9-10)**:1441–1455 DOI 10.1130/B26341.1.

**Retallack GJ. 2013.** Permian and Triassic greenhouse crises. *Gondwana Research* **24(1)**:90–103 DOI 10.1016/j.gr.2012.03.003.

**Retallack GJ, Jahren AH. 2008.** Methane release from igneous intrusion of coal during Late Permian extinction events. *The Journal of Geology* **116**:1–20 DOI 10.1086/524120.

**Retallack GJ, Sheldon ND, Carr PF, Fanning M, Thompson CA, Williams ML, Jones BG, Hutton A. 2011.** Multiple Early Triassic greenhouse crises impeded recovery

from Late Permian mass extinction. *Palaeogeography, Palaeoclimatology, Palaeoecology* **308**(1–2):233–251 DOI 10.1016/j.palaeo.2010.09.022.

**Romano M, Bernardi M, Petti FM, Rubidge B, Hancox J, Benton MJ. 2020.** Early Triassic terrestrial tetrapod fauna: a review. *Earth-Science Reviews* **210**:103331 DOI 10.1016/j.earscirev.2020.103331.

**Romano M, Citton P, Nicosia U. 2015.** Corroborating trackmaker identification through footprint functional analysis: the case study of *Ichniotherium* and *Dimetropus*. *Lethaia* **49**:102–116.

**Romer AS. 1971.** The Chanares (Argentina) Triassic reptile fauna XI. Two new long-snouted thecodonts, Chanaresuchus and *Gualosuchus*. *Breviora* **379**:1–22.

**Santi G, Lualdi A, Decarlis A, Nicosia U, Ronchi A. 2015.** Chirotheriid footprints from the Lower-Middle Triassic of the Briançonnais Domain (Pelite di Case Valmarenca, Western Liguria, NW Italy). *Bollettino della Società Paleontologica Italiana* **54**(2):82.

**Scheyer TM, Romano C, Jenks J, Bucher H. 2014.** Early Triassic marine biotic recovery: the predators' perspective. *PLOS ONE* **9**:e88987 DOI 10.1371/journal.pone.0088987.

**Schmid SM, Fügenschuh B, Kissling E, Schuster R. 2004.** Tectonic map and over-all architecture of the Alpine orogen. *Eclogae Geologicae Helvetiae* **97**:93–117 DOI 10.1007/s00015-004-1113-x.

**Schmid SM, Kissling E, Diehl T, Van Hinsbergen DJJ, Molli G. 2017.** Ivrea mantle wedge, arc of the Western Alps, and kinematic evolution of the Alps—Apennines orogenic system. *Swiss Journal of Geosciences* **110**:581–612 DOI 10.1007/s00015-016-0237-0.

**Schobben M, Joachimski MM, Korn D, Leda L, Korte C. 2014.** Palaeotethys seawater temperature rise and an intensified hydrological cycle following the end–Permian mass extinction. *Gondwana Research* **26**:675–683 DOI 10.1016/j.gr.2013.07.019.

**Schoch RR, Sues HD. 2014.** A new archosauriform reptile from the Middle Triassic (Ladinian) of Germany. *Journal of Systematic Palaeontology* **12**(1):113–131 DOI 10.1080/14772019.2013.781066.

**Sephton MA, Looy CV, Brinkhuis H, Wignall PB, De Leeuw JW, Visscher H. 2005.** Catastrophic soil erosion during the end–Permian biotic crisis. *Geology* **33**:941–944 DOI 10.1130/G21784.1.

**Sepkoski Jr JJ. 1984.** A kinetic model of Phanerozoic taxonomic diversity. III. Post–Paleozoic families and mass extinctions. *Paleobiology* **10**:246–267 DOI 10.1017/S0094837300008186.

**Sereno PC. 1991.** Basal archosaurs: phylogenetic relationship and functional implications. *Journal of Vertebrate Paleontology* **11**:1–53.

**Shen J, Chen J, Algeo TJ, Yuan S, Feng Q, Yu J, Zhou L, O'Connell B, Planavsky NJ. 2019.** Evidence for a prolonged Permian–Triassic extinction interval from global marine mercury records. *Nature Communications* **10**(1):1–9 DOI 10.1038/s41467-019-09620-0.

**Smith RMH, Evans SE. 1996.** New material of *Youngina*: evidence of juvenile aggregation in Permian diapsid reptiles. *Palaeontology* **39**:289–303.

**Song H, Wignall PB, Tong J, Song H, Chen J, Chu D, Tian L, Luo M, Zong K, Chen Y, Lai X. 2015.** Integrated Sr isotope variations and global environmental changes through the Late Permian to early Late Triassic. *Earth and Planetary Science Letters* **424**:140–147 DOI 10.1016/j.epsl.2015.05.035.

**Song H, Wignall PB, Tong J, Yin H. 2013.** Two pulses of extinction during the Permian–Triassic crisis. *Nature Geoscience* **6(1)**:52–56 DOI 10.1038/ngeo1649.

**Sookias RB, Butler RJ. 2013.** Euparkeriidae. *Geological Society, London, Special Publications* **379**:35–48 DOI 10.1144/SP379.6.

**Sues HD, Desojo JB, Ezcurra MD. 2013.** Doswelliidae: a clade of unusual armoured archosauriforms from the Middle and Late Triassic. *Geological Society, London, Special Publications* **379**:SP379–13.

**Sun Y, Joachimski M, Wignall PB, Yan C, Chen Y, Jiang H, Wang L, Lai X. 2012.** Lethally hot temperatures during the early Triassic Greenhouse. *Science* **338**:1–35 DOI 10.1126/science.338.6103.1.

**Torsvik TH, Van Der Voo R, Preeden U, Mac C, Steinberger B, Doubrovine PV, Van Hinsbergen DJJ, Domeier M, Gaina C, Tohver E, Meert JG, McCausland PJA, Cocks LRM. 2012.** Earth-Science Reviews Phanerozoic polar wander, palaeogeography and dynamics. *Earth Science Reviews* **114(3–4)**:325–368 DOI 10.1016/j.earscirev.2012.06.007.

**Treasise G, Sarjeant WAS. 1997.** *The tracks of Triassic vertebrates. Fossil evidence from North-West England.* London: The stationery Office, 204 pp.

**Trotteyn MJ, Arcucci AB, Raugust T. 2013.** Proterochampsia: an endemic archosauriform clade from South America. In: Nesbitt SJ, Desojo JB, Irmis RB, eds. *Anatomy, phylogeny and palaeobiology of early archosaurs and their kin.* 379. London: Geological Society, London, Special Publications, 59–90.

**Van Hinsbergen DJJ, De Groot LV, Van Schaik SJ, Spakman W, Bijl PK, Sluijs A, Langereis CG, Brinkhuis H. 2015.** A paleolatitude calculator for paleoclimate studies. *PLOS ONE* **10(6)**:1–21.

**Vanossi M. 1969.** La serie brianzonese di Salto del Lupo (Liguria Occ.): osservazioni sedimentologico-stratigrafiche. *Atti Ist. Geol. Univ. Pavia* **20**:3–16.

**Vanossi M. 1974.** *L'Unità di Ormea: una chiave per l'interpretazione del Brianzonese ligure. Tipografia del libro.*.

**Vanossi M. 1991.** *Guide geologiche regionali, 11 itinerari, Alpi Liguri (a cura della SGI).* Milano: BE-MA Edit, 296 pp.

**Von Huene F. 1902.** Übersichtüber die Reptilien der Trias. *Geologische und Paläontologische Abhandlungen* **10**:1–84.

**Welles SP. 1947.** Vertebrates from the upper Moenkopi formation of Northern Arizona. *University of California Publications in Geological Science* **27**:241–294.

**Whiteside JH, Ward PD. 2011.** Ammonoid diversity and disparity track episodes of chaotic carbon cycling during the early Mesozoic. *Geology* **39**:99–102 DOI 10.1130/G31401.1.

**Wignall PR. 2001.** Large igneous provinces and mass extinctions. *Earth-Science Reviews* **53**:1–33 DOI 10.1016/S0012-8252(00)00037-4.

**Wilson JA. 2005.** Integrating ichnofossil and body fossil records to estimate locomotor posture and spatiotemporal distribution of early sauropod dinosaurs: a stratocladistic approach. *Paleobiology* **31(3)**:400–423 DOI 10.1666/0094-8373(2005)031[0400:IIABFR]2.0.CO;2.

**Withers PC. 1992.** *Comparative animal physiology.* New York: Saunders College.

**Xing LD, Klein H, Lockley MG, Li J, Zhang J, Matsukawa M, Xiao J. 2013.** Chirotherium trackways from the Middle Triassic of Guizhou, China. *Ichnos* **20**:99–107 DOI 10.1080/10420940.2013.788505.

**Young CC. 1964.** The pseudosuchians in China. *Palaeontologia Sinica Series C* **19**:105–205.

**Zhang F. 1975.** A new thecodont *Lotosaurus*, from Middle Triassic of Hunan. *Vertebrata PalAsiatica* **13**:144–147.

**Ziegler PA, Stampfli GM. 2001.** Late Palaeozoic–Early Mesozoic plate boundary reorganization: collapse of the Variscan orogen and opening of Neotethys. In: Cassinis G, ed. *Permian continental deposits of europe and other areas. Regional reports and correlations.* 25. Brescia: Annali Museo Civico Scienze Naturali, 17–34.