# Peer review of "Archosauriform footprints in the Lower Triassic of Western Alps and their role in understanding the effects of the Permian-Triassic hyperthermal"

_PeerJ, doi:10.7717/peerj.10522_

## Round 0.1 · original submission · Minor Revisions

Dear authors,

We have now three review reports about your manuscript on “Archosauriform footprints in the Lower Triassic of Western Alps” and all of them have agreed that it is an interesting contribution and would require only minor modifications to be acceptable for publishing in PeeJ.

Commentaries from the reviewers are critically constructive and I consider they will improve your already well constructed and interesting manuscript. I recommend that you follow all the suggestions that they included in the annotated pdfs provided.

I also would like you to revise the following paragraph at lines 511 to 518:

“Early Triassic erythrosuchid skeletal fossils are known from the late Olenekian of Russia, South Africa, China and India (see Gower, 2003; Ezcurra et al., 2013, 2019, 2020; Gower et al., 2014; Ezcurra, 2016). The Gardetta ichnosite testifies the presence of erythrosuchids and more generally of Archosauriformes at low latitudes (11.8° N) also during the Early Triassic (Fig. 11). This supports the conclusions of Bernardi et al. (2015, 2018) that Early Triassic ichnosites are mainly distributed along the tropics, contrasting the pattern described by skeletal findings and the hypothesis of a low-latitude vacancy of continental tetrapods during or soon after the PTME (Sun et al., 2012).”

I would like you to take into account that many of these Early Triassic sites in Russia, South Africa and China have yielded erythrosuchid skeletal remains and also footprints assigned to that group. Thus, it will be interesting if you provide a discussion about the taphonomic bias that favored the preservation of only tracks in the Gardetta site. Probably, this request is allied to the reviewers concerns related to the inconvenience to match some tracks to a particular trackmaker when you cannot demonstrate it with a reasonable degree of confidence. So, please, leave an “open door” to other eventual possibilities, you can suggest that it may be probable that the tracks were produced by erythrosuchids, but this cannot be assured unless until future findings (perhaps including skeletal fossils) provide additional, more confident evidence.

This observation also applies to your conclusion that low latitude tetrapod communities were not vacated by the Permo-Triassic extinction or that they experienced a rapid recovery. There is a possibility that the findings you describe here are casual and thus do not represent for now, a guarantee for refuting previous suggestions. Thus, further studies will be required to arrive to more confident conclusions, but this is a promising starting.

I hope you find useful all the comments and suggestions and you can submit the revised version of your manuscript very soon.

With my best regards,
Graciela Piñeiro

·

Basic reporting

No comment (see note to editor).

Experimental design

No comment (see note to editor).

Validity of the findings

No comment (see note to editor).

Additional comments

Porto Alegre, August 20, 2020

PeerJ
Dear authors

REFEREE REPORT

The manuscript entitled “Archosauriform footprints in the Lower Triassic of Western Alps and their role in understanding the effects of the Permian-Triassic hyperthermal” (ID: 51782) and authored by Fabio M. Petti, Heinz Furrer, Enrico Collo, Edoardo Martinetto, Massimo Bernardi, Massimo Delfino, Marco Romano and Michele Piazza was completely revised by me.
In this manuscript, the authors described a new occurrence of fossil tracks (Chirotherium and Isochirotherium) assigned to archosauriforms from the Lower Triassic deposits of the Gardetta Plateau of NW Italy. A new ichnospecies of Isochirotherium was erected (I. gardettae isp. nov.) and a discussion on the distribution of tetrapods on low latitudes during the Early Triassic was made.
The length and structure of the manuscript is good, as well the number (11) and quality of the figures. The authors cited both classic and up-to-date papers on the manuscript. Because I am not an English native-speaker, I was not able to evaluate the idiom, but the manuscript seems to be very coherent and readable. Other punctual suggestions and corrections were made directly on the manuscript file that I send to you.
The only point I think can be better explored is the differences on manual dactyly between the Isochirotherium ichnospecies. The authors used this character to differentiate the materials and erect the new isp. However, there are no comments made on a potential preservation bias on it, because it is not unlikely pentadactyl tetrapod can produce tetra- or tridactyl manual footprints due the substrate conditions and variations on the weigh center and hand kinematics of the producers (see, for example, the works of Peter Falkingham on the “Goldilocks Effect”).
Based on what I mentioned above, I consider that the manuscript likely publishable in PeerJ after pass by minor modifications (minor revision request).
I am available for any clarification you may want.

Sincerely yours,
Dr. Heitor Francischini
Universidade Federal do Rio Grande do Sul
heitorfrancischini@hotmail.com / heitor.francischini@ufrgs.br
ORCID: 0000-0001-9809-7784

·

Basic reporting

The English grammar used through the manuscript is clear and professional.
The background is well presented and completed with ichnological, taxonomical, and geological actualized literature. I have not cross-checked the reference list.
The figures and tables are well executed and are all necessary to support the results presented in the paper. The structure is correct for this kind of paper.

Experimental design

The research is original, with analysis of primary and bibliographic data.
The technical methods used to analyze the data are modern, adequate, and well described for replication.
Research questions are clearly stated, relevant, and meaningful, and are followed throughout the manuscript.

Validity of the findings

The manuscript is a good example of how to analyze in detail the ichnological record and used this information for an ichnotaxonomical, taxonomical, paleobiogeografical and biochronological studies.
The results are robust and the conclusions relevant.

Additional comments

The authors propose a compressive manuscript about the relationship among the Early Triassic ichnofauna found in the North of Italy, the Permo-Triassic Mass Extinction, and the paleobiogeographical bias. Based on geologic and biochronologic information, they pointed out that vertebrate tracks are probably late Olenekian in age. The authors concluded that the ichnological record is a useful tool to study the presence of archosauriformes at low latitudes during the Early Triassic epoch.
I have added some comments in the attached pdf about the ichnotaxonomy of Isochirotherium and the choice of useful ichnotaxobases.
On the other hand, I wonder why the bone vertebrate record is scarce in the Early Triassic in the equatorial zone, but there are abundant vertebrate tracks. Maybe taphonomical biases?

·

Basic reporting

The manuscript is quite well prepared with excellent illustrations. However, there are many small errors of English grammar: see edited PDF. Please make agreement between plural and singular subjects and verbs.
I have doubts about the validity of erecting a new ichnospecies of Isochirotherium because the new ichnospecies is only briefly compared with I. soergeli (the type). Ideally the new ichnospecies should have a clear comparative diagnosis. Authors should explain how they determine whether the tetradactyl manus is a preservational (extramorphological) phenomenon, or evidence of front limb posture. The authors "open the door" to discussing this problem after ichnotaxonomy section but do not adequately relate problem 'back to' ichnotaxonomy afterwards.

There is also the problem that the quality of preservation is not addressed. Recently it has become fashionable to refer to the quality of preservation scale of Belvedere and Farlow (2016) and Marchetti et al. (2019). These scales do not need to be followed religiously, but they should be mentioned. This is easily done

Experimental design

no comment

Validity of the findings

The use of the term "trackmaker identification" should be introduced more cautiously. It is rarely possible to identify the trackmaker precisely at the species or genus level. SO THE TRUTH IS THIS IS AN EXERCISE IN "INFERRED TRACKMAKER IDENTIFICATION." The paper proves this by suggesting that Chirotherium and Isochirotherium can only tentatively be "correlated" with trackmakers at the family level. There are 7-8 named Isochirotherium ichnospecies, and we cannot match each with a trackmaker taxon with any confidence. I recommend that the authors make this uncertainly clear. (However, ichnotaxa are clearly useful in providing paleoecological information, especially when body fossils are incomplete, for example with no feet. !!

The discussion and conclusions are a little speculative, but acceptable because the ideas are interesting and provocative and it seems the PTME and aftermath have become a subject of repeated discussion by many authors.

OK

Additional comments

Interesting paper. Minor to very moderate revision suggested
Please pay attention to my comments about:
English grammar
meaning of "inferred trackmaker identification"
"Quality of preservation"
and rigorous criteria for naming new ichnospecies

---

## Round 0.2 · Minor Revisions

Dear authors,
Thanks for have modified your manuscript according to the suggestions and comments that you received from reviewers and editors.

A new reconsideration of the revised manuscript was now completed and we acknowledge with the reviewer that your work would be ready for publication in PeerJ. However, taking into account that the trackway of the described new ichnospecies had to be left in the field and the solitary footprints are displaced from their original strata and thus decontextualized from the study area, we have decided, along with the PeerJ Editorial Staff, that before final acceptation for publication you should provide a 3D photogrammetry model that can be published at a public repository (e.g. Morphosource) in order to facilitate that the new ichnospecies can be used for future studies on the area.
With my best regards,
Graciela Piñeiro

·

Basic reporting

OK

Experimental design

OK

Validity of the findings

OK

Additional comments

After reviewing this paper a second time, and the rebuttal comments, I think it can be published without further revision. However, I defer to the editors and other reviewers to see if they have any further suggestions.
I note however that I was not the only reviewer to ask whether the trackmaker had a truly tetradactyl manus and not a pentadactyl manus that was incompletely preserved.
This will be something to consider in future studies. Chirotheres are sometimes categorized as small manus and large manus.

---

## Round 0.3 · Minor Revisions

Dear authors,


I apologize for the delay. We are very close to the acceptation of your article for to be published in PeerJ. However I have read again the text at the Material and Methods section and I realized that the specimens GD-E1, GD-E2, GD-12 were not assigned to any of the described ichnotaxa. From my last revision I could get the following information:

Chirotherium isp.: (GT-1 and GT-2), two trackways preserved as concave epirelief.
Apparently they were left in the field because they are in large slabs.
GD-E1, GD-E2, GD-12 (not assigned to an ichnotaxon?): Three solitary small footprints (from loose slabs) preserved as convex epirelief and currently stored at the Museo di Geologia e Paleontologia dell’Università di Torino (Turin, Italy).

Isochirotherium gardettensis ichnosp. nov. Cast MGPT-PU135785 (Museo di Geologia e Paleontologia dell’Università di Torino, Italy) printed after the 3D-modelling of the GT-7 and GT-3 manus-pes couple.
Referred specimens: GT-7, a trackway made of three exceptionally-preserved and consecutive manus-pes couples not exceeding 2.20 m across (left in the field). GT-3, isolated, partially preserved track (stored at the Museum di Geologia e Paleontologia dell’Università di Torino (Turin, Italy).

Thus, it will be better that you provide in a Table the taxonomic affinities of each of the studied specimens cited in the text and their repository, remarking the print/s that was/were used for the 3D model uploaded in Morphospace.
Kind regards,

Graciela Piñeiro

---

## Round 0.4 · Minor Revisions

Dear authors,

Okay, but I need that you specify where these isolated footprints are deposited as well as all the described materials. Thus, please, provide a Table including specimen number, taxonomic affinities and repository, remarking the print/s that was/were used for the 3D model uploaded in Morphospace. In that way, all the required information will be of easier access. It will take you just a few minutes.
Kind regards,

Graciela Piñeiro

---

## Round 0.5 · accepted · Accept

Dear authors,

Many thanks for the changes inserted into the manuscript and to provide the table from which all the required information can be easily accessed.

From my part, I consider that all the changes and additions requested have been successfully completed to improve the manuscript, and I have no other concerns. Therefore, your manuscript is acceptable to be published in PeerJ. Congratulations!

Best wishes,

Graciela Piñeiro